# Learning Label Refinement and Thresholds for Imbalanced Semi-Supervised Learning

## Abstract

Semi-supervised learning (SSL) has proven to be effective in enhancing generalization when working with limited labeled training data. Existing SSL algorithms based on pseudo-labels rely on heuristic strategies or uncalibrated model confidence and are unreliable when imbalanced class distributions bias pseudo-labels. In this paper, we introduce SEmi-supervised learning with pseudo-label optimization based on VALidation data (SEVAL) to reduce the class bias and enhance the quality of pseudo-labelling for imbalanced SSL. First, we develop a curriculum for adjusting logits, improving the accuracy of the pseudo-labels generated by biased models. Second, we establish a curriculum for class-specific thresholds, ensuring the correctness of pseudo-labels on a per-class basis. Importantly, SEVAL adapts to specific tasks by learning refinement and thresholding parameters from a partition of the training dataset in a class balanced way. Our experiments show that SEVAL surpasses current methods based on pseudo-label refinement and threshold adjustment, delivering more accurate and effective pseudo-labels in various imbalanced SSL situations. Owing to its simplicity and flexibility, SEVAL can readily be incorporated to boost the efficacy of numerous other SSL techniques.

## 1 Introduction

Semi-supervised learning (SSL) algorithms are trained on datasets that contain both labelled and unlabelled samples Chapelle et al. (2009). SSL improves representation learning and refines decision boundaries without relying on large volumes of labeled data, which are labor-intensive to collect.

Numerous SSL algorithms have been introduced, with one of the most prevalent assumptions being entropy minimization, which requires the decision boundaries to lie in low density areas Wang et al. (2022a). In order to achieve this, pseudo-labels are introduced in the context of SSL Scudder (1965), and this concept has been extended to numerous variants, including recent developments Laine & Aila (2016); Berthelot et al. (2019b;a); Sohn et al. (2020); Zhang et al. (2021); Wang et al. (2022b). In the pseudo-label framework, models trained with labelled data periodically classify the unlabelled samples and samples that are confidently classified are incorporated into the training set.

The success of pseudo-label based SSL algorithms hinges on the quality of the pseudo-labels Chen et al. (2023). Nevertheless, when implemented in real-world applications, the performance of these SSL algorithms often experiences a significant degradation due to the prevalence of class imbalance in real-world datasets Liu et al. (2019). In particular, when exposed to imbalanced training data, the model tends to become sensitive to the majority class Cao et al. (2019); Li et al. (2020). Consequently, this sensitivity impacts the generated pseudo-labels, introducing a bias in the process.

In this paper, we propose SEmi-supervised learning with pseudo-label optimization based on VALidation data (SEVAL), a learning strategy aimed at enhancing the performance of pseudo-label based SSL algorithms when trained on imbalanced training datasets. We break down the designs of predominant imbalanced SSL algorithms into components, and introduce substantial enhancements to various components, substantiated by detailed experiments and analysis. Specifically, SEVAL refines the decision boundaries of pseudo-labels by learning a curriculum for the logit offsets. The optimization process of SEVAL closely resembles that of AutoML, as both involve the learning of a set of hyper-parameters from a partition of the training dataset before proceeding with the standard training process Zoph & Le (2016); Ho et al. (2019). In this way, SEVAL can adapt to the specific

task by learning from the imbalanced data itself, resulting in a better fit. Moreover, SEVAL optimizes confidence thresholds to select pseudo-labels that are fair to different classes. The learned thresholds can effectively prioritize the selection of samples from the high-precision class, a common occurrence in imbalanced SSL but typically overlooked by current model confidence-based dynamic threshold solutions Zhang et al. (2021); Guo & Li (2022).

The contributions of this paper are as follow:

- We propose to establish a curriculum of pseudo-label adjustment offsets to reduce the class bias of pseudo-labels for imbalanced SSL algorithms. It can be viewed as an enhanced extension of heuristic post-hoc logit adjustment techniques, better suited to underlying tasks and delivering improved accuracy in both pseudo-labeling and inference.

- We propose to learn a curriculum of thresholds to select confidently classified pseudo-labels based on a labelled validation dataset using a novel optimization function. The obtained thresholds notably improve the performance of the minority class, accommodating all four threshold adjustment scenarios, whereas existing methods falter in two out of the four.

- We combine the two techniques into a learning framework, SEVAL, and find that it can outperform state-of-the-art pseudo-label based methods under various imbalanced SSL scenarios. SEVAL does not demand any supplementary computation after the curricula are acquired and offers flexibility for integration into other SSL algorithms.

## 2 RELATED WORK

**Semi-supervised learning.** SSL has been a longstanding research focus. The majority of SSL approaches have been developed under the assumption of consistency, wherein samples with similar features are expected to exhibit proximity in the label space Chapelle et al. (2009); Zhou et al. (2003). Apart from graph-based methods Iscen et al. (2019); Kamnitsas et al. (2018), perturbation-based methods Xie et al. (2020); Miyato et al. (2018) and generative model-based methods Li et al. (2017); Gong et al. (2023), a more straightforward solution is using pseudo-labels to periodically learn from the model itself to encourage entropy minimization Grandvalet & Bengio (2004).

Deep neural networks are particularly suited for pseudo-label-based approaches due to their strong classification accuracy, enabling them to generate high-quality pseudo-labels Lee et al. (2013); Van Engelen & Hoos (2020). Several methods have been explored to generate pseudo-labels with a high level of accuracy Wang et al. (2022a); Xu et al. (2021). For example, Mean-Teacher Tarvainen & Valpola (2017) calculates the pseudo-label using the output of an exponential moving average (EMA) model along the training iterations; MixMatch Berthelot et al. (2019b) derives pseudo-labels by averaging the model predictions across various transformed versions of the same sample; FixMatch Sohn et al. (2020) estimates pseudo-labels of a strongly augmented sample with the model confidence on its weakly augmented version; Built upon FixMatch, FlexMatch and FreeMatch Zhang et al. (2021); Wang et al. (2022b) choose confidently classified samples based on the model's learning progress, which results in the selection of more samples if the model is not learning effectively. SEVAL can seamlessly adapt current pseudo-label based SSL algorithms to real world application by tackling the class imbalance bias of pseudo-labels.

**Imbalanced semi-supervised learning.** The potential and practical implications of SSL have captured the attention of numerous research studies. There are mainly three groups of methods to tackle the challenge of class imbalance in SSL. The first group of methods alters the cost function computed using the labeled samples to train a balanced classifier, consequently leading to improved pseudo-labels. The research on long-tailed recognition, which focuses on building balanced classifiers through adjusted cost functions or model structures in a completely supervised learning environment, frequently inspires those works Chawla et al. (2002); Kang et al. (2019); Menon et al. (2020); Zhang et al. (2023); Tian et al. (2020). BiS He et al. (2021) and SimiS Chen et al. (2022) resample the labelled and pseudo-labelled training datasets to build balanced classifier. ABC decouples the feature learning and classifier learning with a two head model architecture Lee et al. (2021). SAW reweights unlabeled samples from different classes based on the learning difficulties Lai et al. (2022b). The second category of methods refines the pseudo-labels to achieve a balanced distribution across classes. DARP Kim et al. (2020) refines pseudo-labels by aligning their distribution with the target distribution. SaR Lai et al. (2022a) aligns pseudo-labels to true distributions us-

ing distribution alignment (DA)-based mitigation vector. Adsh Guo & Li (2022) utilizes adaptive threshold to ensure that similar number of pseudo-labels are selected for each class. Finally, some hybrid methods simultaneously adjust the cost functions and refine the pseudo-labels. For instance, apart from bootstrap sampling strategy, CReST+ Wei et al. (2021) utilize DA to adjust the class bias of pseudo-labels. DASO Oh et al. (2022) improves pseudo-labels with semantic pseudo-labels and regularizes the feature encoder by aligning balanced semantic prototypes. ACR Wei & Gan (2023) is a holistic approach that builds upon the successes of ABC, FixMatch and MixMatch, and utilizes logit adjustment (LA) to refine pseudo-labels Menon et al. (2020), yielding impressive results.

SEVAL seamlessly integrates into SSL pipelines without necessitating alterations to the model architecture, data sampling process, or additional pseudo-label calculations. In addition, unlike many imbalanced SSL algorithms such as Adsh, DARP and CreST+, SEVAL does not make any assumptions on the distribution of unlabelled data, thus it can be applied to scenarios where the distributions of labelled and unlabelled data are distinct without any modifications.

## 3 PRELIMINARIES

We consider the problem of $C$-class imbalanced semi-supervised classification. Let $X \subset \mathbb{R}^d$ be the feature space and $Y = \{1, 2, \ldots, C\}$ be the label space. For a labelled training dataset $\mathcal{X} = \{(\boldsymbol{x}_i, y_i)\}_{i=1}^N$ with a total of $N$ labelled samples, where each $(\boldsymbol{x}_i, y_i) \in (X \times Y)$, the class distribution is imbalanced, with varying numbers of samples per class, denoted $n_c$. Assuming $\boldsymbol{n}$ is a vector that contains $n_c$ for different class $c$ in a descending order, we define the imbalance ratio $\gamma$ as $\gamma = \max_j(n_j) / \min_j(n_j)$ (typically exceeds 10). We also have access to $M$ unlabelled samples, represented as $\mathcal{U} = \{\boldsymbol{u}_i\}_{i=1}^M$, which contain $m_c$ samples for class $c$. After optimization, we expect the model perform well on a separate test dataset $\mathcal{T}$ which have uniform class distributions.

A model $f$ is a function that produces the class conditionals $P_{\mathcal{X}}(y|\boldsymbol{x}) = \boldsymbol{p}_i^{\mathcal{X}} \in \mathbb{R}^C$ given a labelled sample $\boldsymbol{x}_i$, with its $c$'th element $p_{ic}^{\mathcal{X}} \in [0, 1]$ corresponding to the $c$'th class. The predicted probability $\boldsymbol{p}_i^{\mathcal{X}}$ is obtained by applying the softmax function to the network output $\boldsymbol{z}_i^{\mathcal{X}} = f(\boldsymbol{x}_i)$ such that $p_{ic}^{\mathcal{X}} = \sigma(\boldsymbol{z}_i^{\mathcal{X}})_c = \frac{e^{z_{ic}^{\mathcal{X}}}}{\sum_{j=1}^C e^{z_{ij}^{\mathcal{X}}}}$. The model $f$ is commonly optimized by minimizing $\mathcal{L}_{\text{cls}} = \frac{1}{N} \sum_{i=1}^N \mathcal{H}(y_i, \boldsymbol{p}_i^{\mathcal{X}})$ in the supervised learning setting, where $\mathcal{H}$ is the cross-entropy loss.

In order to optimize with unlabelled data, pseudo-labeling techniques are commonly adopted to regularize the network parameters by learning from the model itself Lee et al. (2013). Rather than relying on the actual ground truth label, we generate a pseudo-label probability vector $\boldsymbol{q}_i \in \mathbb{R}^C$ for an unlabelled sample $\boldsymbol{u}_i$. The pseudo-label $\hat{y}_i$ is then determined as $\arg\max_j q_{ij}$. Note that, here we describe the case of hard pseudo-label for simplicity, but the method generalizes to the case of soft pseudo-label. With a model prediction $\boldsymbol{p}_i^{\mathcal{U}} = f(\boldsymbol{u}_i)$, the model is optimized to minimize:

$$\mathcal{L}_{\text{u}} = \frac{1}{M} \sum_{i=1}^M \mathbb{1}(\max_j(q_{ij}) \geq \tau) \mathcal{H}(\hat{y}_i, \boldsymbol{p}_i^{\mathcal{U}}), \tag{1}$$

where $\mathbb{1}$ is the indicator function, and $\tau$ is a predefined threshold that filters out pseudo-labels with low confidence. Generating pseudo-labels constitutes a crucial stage in the implementation of semi-supervised learning algorithms Laine & Aila (2016); Sohn et al. (2020); Berthelot et al. (2019b;a). Specifically, FixMatch Sohn et al. (2020) produces the pseudo-label of a strongly-augmented (i.e. RandAugment Cubuk et al. (2020)) version $\mathcal{A}_s(\boldsymbol{u}_i)$ based on the model prediction of its weakly-augmented (i.e. flipped and shifted) copy $\mathcal{A}_w(\boldsymbol{u}_i)$. Specifically, the semi-supervised algorithm is optimized with $\mathcal{A}_s(\boldsymbol{u}_i)$ using the pseudo-label probability calculated as $\boldsymbol{q}_i = \sigma(f(\mathcal{A}_w(\boldsymbol{u}_i)))$. Given its simplicity and strong performance, we employ FixMatch as our primary baseline for the majority of experiments conducted in this study.

When trained with imbalanced training data $\mathcal{X}$, the model $f$ will be biased at inference time. Therefore, in this case the generated pseudo-labels probability $\boldsymbol{q}_i$ by common SSL algorithms would become more sensitive to the majority class and make the model bias even worse.

In this study, we focus on the method to refine the $\boldsymbol{q}_i$ under this circumstance. At the same time, we expand the threshold to operate on a class-specific basis and acquiring a set of $\boldsymbol{\tau} \in \mathbb{R}^C$ values to achieve accuracy fairness. The model can then dynamically select the appropriate thresholds based on its prediction. In the following section, we will bypass the computation of pseudo-label probability $\boldsymbol{q}_i$ and concentrate on our contributions.

# 4 SEVAL

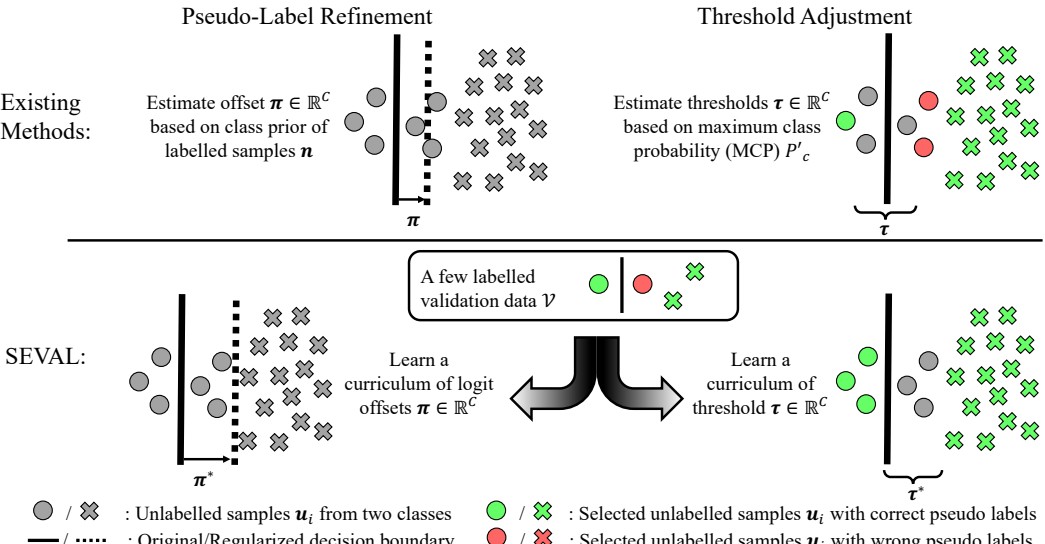

Figure 1: Overview of SEVAL optimization process which consists of two learning strategies aiming at mitigating bias in pseudo-labels within imbalanced SSL scenarios: 1) Pseudo-label refinement and 2) Threshold Adjustment. The curriculum for parameter learning is determined through the evaluation of validation data performance, ensuring greater accuracy while preventing overfitting.

Figure 1 shows an overview of SEVAL. It comprises two optimization processes including pseudo-label refinement and threshold adjustment. Importantly, we propose to optimize these parameters using a separate labelled validation dataset. Independent of the training dataset $\mathcal{X}$ and $\mathcal{U}$, we assume we have access to a validation dataset $\mathcal{V} = \{(\boldsymbol{x}_i, \boldsymbol{y}_i)\}_{i=1}^{K}$, which contains $k_c$ samples for class $c$. We make no assumptions regarding $k_c$; that is, $\mathcal{V}$ can either be balanced or imbalanced. The method is presented in details in the following sections.

## 4.1 LEARNING PSEUDO-LABEL REFINEMENT

For an unlabeled sample $\boldsymbol{u}_i$, we determine its pseudo-label probability $\boldsymbol{q}_i$ based on its corresponding pseudo-label logit $\hat{\boldsymbol{z}}_i^{\mathcal{U}}$. In the process of pseudo-label refinement, we aim to adjust the decision boundaries for $\hat{\boldsymbol{z}}_i^{\mathcal{U}}$ with offset $\boldsymbol{\pi} \in \mathbb{R}^C$ to reduce class biases.

Here derive the theoretical optimal thresholds based on Bayes theorem. Given that the test distribution $\mathcal{T}$ shares identical class conditionals with the training dataset $\mathcal{X}$ (i.e., $P_{\mathcal{X}}(X|Y) = P_{\mathcal{T}}(X|Y)$) and deviates solely in terms of class priors ($P_{\mathcal{X}}(Y) \neq P_{\mathcal{T}}(Y)$), we can assert:

**Theorem 1** *Given that a Bayes classifier $f^*(y|\boldsymbol{x})$ is optimized on $P_{\mathcal{X}}(X, Y)$,*

$$f_{\mathcal{T}}(y|\boldsymbol{x}) = \frac{f^*(y|\boldsymbol{x})P_{\mathcal{T}}(y)}{P_{\mathcal{X}}(y)}, \tag{2}$$

*is the optimal Bayes classifier on $P_{\mathcal{T}}(X, Y)$, where $P_{\mathcal{X}}(X|Y) = P_{\mathcal{T}}(X|Y)$ and $P_{\mathcal{X}}(Y) \neq P_{\mathcal{T}}(Y)$.*

**Corollary 1.1** *The Bayes classifier $f_{\mathcal{U}}(y|\boldsymbol{x}) = f_{\mathcal{T}}(y|\boldsymbol{x})$ should be also optimal on the resampled validation dataset $\frac{P_{\mathcal{U}}(X, Y)P_{\mathcal{T}}(Y)}{P_{\mathcal{U}}(Y)}$, where $P_{\mathcal{T}}(X|Y) = P_{\mathcal{U}}(X|Y)$ and $P_{\mathcal{T}}(Y) \neq P_{\mathcal{U}}(Y)$.*

The theorem provides insight into the formulation of pseudo-label offsets: it is contingent not on the distribution of unlabeled data, $P_{\mathcal{U}}$, but rather on the distribution of test data, $P_{\mathcal{T}}$. From this analytical viewpoint, we present a summarized Table 1 of current pseudo-label refinement solutions. DA Berthelot et al. (2019a); Wei et al. (2021); Kim et al. (2020) is a commonly employed technique to make balanced prediction for different classes which align the predicted class priors $\tilde{P}_{\mathcal{U}}(Y)$ to true class priors of $\mathcal{U}$, making the model being fair Bridle et al. (1991). It only reduces the calibration errors but cannot be optimal because it does not take $P_{\mathcal{T}}$ into account. LA adjust the network

prediction from $\arg\max_c \hat{z}_{ic}^{\mathcal{U}}$ to $\arg\max_c(\hat{z}_{ic}^{\mathcal{U}} - \beta\log\pi_c)$, where $\beta$ is a hyper-parameter and $\boldsymbol{\pi}$ is determined as the empirical class frequency Menon et al. (2020); Zhou & Liu (2005); Lazarow et al. (2023). It shares similar design with Eq. 1, however, recall that theorem 1 provides a justification for employing logit thresholding when optimal probabilities $f^*(y|\boldsymbol{x})$ are accessible. Although neural networks strive to mimic these probabilities, it is not realistic for LA as the classifier is not optimal during training and neural networks are often uncalibrated and over confident Guo et al. (2017).

| | DA | LA | DASO | SEVAL |
|---|---|---|---|---|
| Estimation of $f_{\mathcal{U}}(y|\boldsymbol{x})$ | $\dfrac{f(y|\boldsymbol{x})P_{\mathcal{U}}(y)}{\tilde{P}_{\mathcal{U}}(y)}$ | $\dfrac{f(y|\boldsymbol{x})P_{\mathcal{T}}(y)}{P_{\mathcal{X}}(y)}$ | Blending similarity based pseudo-label | $\dfrac{f(y|\boldsymbol{x})P_{\mathcal{T}}(y)}{\boldsymbol{\pi}^*}$ |
| Note | Ignoring $P_{\mathcal{T}}(Y)$, thus failing when $P_{\mathcal{T}}(Y) \neq P_{\mathcal{U}}(Y)$. | Inaccurate as $f$ is suboptimal and uncalibrated. | Relying on the effectiveness of blending strategies. | Optimizing the decision boundary on $\mathcal{U}$ using $\mathcal{V}$ as a proxy without assuming a specific $f$. |

Table 1: Theoretical comparisons of SEVAL and other pseudo-label refinement methods including distribution alignment (DA) Berthelot et al. (2019a); Wei et al. (2021); Kim et al. (2020); Lai et al. (2022a), logit adjustment (LA) Wei & Gan (2023); Menon et al. (2020) and DASO Oh et al. (2022).

Therefore, in this study, we further harness its potential by optimizing $\boldsymbol{\pi}$ from the data itself. Assuming the validation data distribution has the same class conditional likelihood as others and $P_{\mathcal{T}}(Y)$ is uniform, SEVAL can directly estimate the optimal decision boundary as required in Theorem 1. Specifically, the optimal offsets $\boldsymbol{\pi}$, are optimized using the labelled validation data $\mathcal{V}$ with:

$$\boldsymbol{\pi}^* = \arg\min_{\boldsymbol{\pi}} \frac{1}{K}\sum_{i=1}^{K}\mathcal{H}(y_i, \boldsymbol{p}_i^{\mathcal{V}}) = \arg\min_{\boldsymbol{\pi}} \frac{1}{K}\sum_{i=1}^{K}\mathcal{H}(y_i, \sigma(\boldsymbol{z}_i^{\mathcal{V}} - \log\boldsymbol{\pi})). \tag{3}$$

Subsequently, we can compute the refined pseudo-label logit as $\hat{z}_i^{\mathcal{U}} - \log\boldsymbol{\pi}^*$, which are expected to become more accurate on a class-wise basis. Of note, we utilize the final learned $\boldsymbol{\pi}^*$ to refine the test results and expect it to perform better than LA.

## 4.2 LEARNING THRESHOLD ADJUSTMENT

Dynamic thresholds have been previously explored in the realm of SSL. Nevertheless, we contend that existing confidence-based threshold methods may falter in two of four scenarios of imbalanced SSL, specifically when a class exhibits high recall and high precision or low recall and low precision.

**Hypothesis 1** *A better thresholds $\boldsymbol{\tau}$ for choosing effective pseudo-labels can be derived from class-specific precision, instead of recall.*

Existing dynamic threshold approaches Zhang et al. (2021); Wang et al. (2022b); Guo & Li (2022) derive the threshold for class $c$ based on the maximum class probability (MCP) of class $c$, i.e. $P_c' = \frac{1}{K_c}\sum_{i=1}^{K} \mathbb{1}_{ic}\max_j p_{ij}^{\mathcal{U}}$, where $\mathbb{1}_{ic} = \mathbb{1}(\arg\max_j(p_{ij}^{\mathcal{V}}) = c)$ is 1 if the predicted most probable class is c an 0 otherwise. The class-wise probability $P_c'$, can be used to estimate the model learning status, or accuracy Guo et al. (2017) (which is equivalent to recall when assessed on a per-class basis since negative samples are not considered) of test samples Garg et al. (2022); Li et al. (2022). Thus, current dynamic methods like FlexMatch also employ it to approximate the threshold for selecting confident pseudo-labels. Nevertheless, it is crucial to recognize that thresholds are not solely reliant on recall. In contrast, as demonstrated in Figure 2, *precision should be the determining factor for thresholds*. While *Case 1* and *Case 2* are the most common scenarios, current MCP-based approaches struggle to estimate thresholds effectively in other situations. We substantiate this assertion in the experimental section, where we find that *Case 3* frequently arises for the minority class in imbalanced SSL and is currently not adequately addressed, as shown in appendix Section D.

However, precision cannot be determined by confidence scores alone, and an external labelled dataset is required. Thus, here we propose a novel strategy to learn the optimal thresholds based on an external validation dataset $\mathcal{V}$. We optimize the thresholds in a manner that ensures the selected samples from different classes achieve the same accuracy level of $t$. This is achieved by:

$$\tau_c^* = \begin{cases} \arg\min_{\tau_c}\left|\frac{1}{s_c}\sum_{i=1}^{K}\mathbb{1}_{ic}\mathbb{1}(y_i = c)\mathbb{1}(\max_j(p_{ij}^{\mathcal{V}}) > \tau_c) - t\right| & \text{if } t < \alpha_c \\ 0 & \text{otherwise} \end{cases}, \tag{4}$$

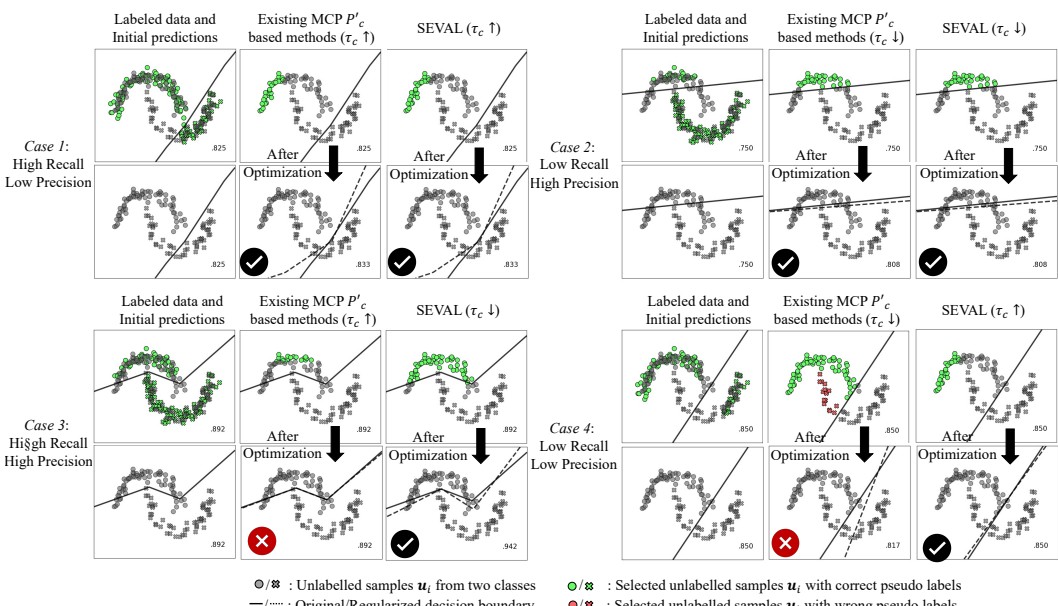

Figure 2: Two-moons toy experiments illustrating the relationship between threshold choice and model performance for class ●. Current MCP-based dynamic thresholding methods such as Flex-Match Zhang et al. (2021), emphasizing recall, may not be reliable for *Case 3* and *Case 4*.

where $s_c = \sum_{i=1}^{K} \mathbb{1}_{ic}\mathbb{1}(\max_j(p_{ij}^{\mathcal{V}}) > \tau_c)$ is the number of samples predicted as class $c$ with confidence larger than $\tau_c$, where $\alpha_c = \frac{1}{K_c}\sum_{i=1}^{K} \mathbb{1}_{ic}\mathbb{1}(y_i = c)$ is the average accuracy of all the samples predicted as class $c$ and $K_c = \sum_{i=1}^{K} \mathbb{1}_{ic}$ is the number of samples predicted as $c$.

Importantly, optimized thresholds are inversely related to precision and possess practical utility in handling classes with varying accuracy. Therefore, we believe this cost function is better suited for fair threshold optimization across diverse class difficulties. In practical scenarios, we often face difficulties in directly determining the threshold through Eq. 4 due to the imbalances in validation data and constraints arising from a limited sample size. To address these issues, we employ normalized cost functions and group-based learning, detailed further in appendix Section C.

After obtaining the optimal refinement parameters, for pseudo-label $\hat{y}_i = \arg\max_j(q_{ij})$ and predicted class $y_i' = \arg\max_j(p_{ij}^{\mathcal{U}})$, we can calculate the unlabelled loss $\mathcal{L}_u = \frac{1}{M}\sum_{i=1}^{M}\mathbb{1}(\max_j(q_{ij}) \geq \tau_{y_i'}^{(l)})\mathcal{H}(\hat{y}_i, \boldsymbol{p}_i^{\mathcal{U}})$ to update our classification model parameters.

### 4.3 CURRICULUM LEARNING

In practice, we learn the curriculum of $\boldsymbol{\pi}$ and $\boldsymbol{\tau}$ based on a partition of labelled training dataset $\mathcal{X}$ thus we do not require additional samples. Specifically, before standard SSL process, we partition $\mathcal{X}$ into two subset $\mathcal{X}'$ and $\mathcal{V}'$ which contain the same number of samples to learn the curriculum.

In order to ensure curriculum stability, we update the parameters with exponential moving average (EMA). Specifically, when we learn a curriculum of length $L$, after several iterations, we optimize $\boldsymbol{\pi}$ and $\boldsymbol{\tau}$ sequentially based on current model status. We then calculate the curriculum for step $l$ as $\boldsymbol{\pi}^{(l)} = \rho_\pi \boldsymbol{\pi}^{(l-1)} + (1 - \rho_\pi)\boldsymbol{\pi}^{(l)*}$ and use this to refine pseudo-label before the next SEVAL parameter update. We provide more implementation details in appendix Section A.

## 5 EXPERIMENTS

We conduct experiments on imbalanced SSL benchmark including CIFAR-10-LT, CIFAR-100-LT Krizhevsky et al. (2009) and STL-10-LT Coates et al. (2011) under the same codebase following Oh et al. (2022). Specifically, we choose wide ResNet-28-2 Zagoruyko & Komodakis (2016) as the feature extractor and train the network at a resolution of $32 \times 32$. We train the neural networks for 250,000 iterations with fixed learning rate of 0.03. We control the imbalance ratios for both labelled

and unlabelled data ($\gamma_l$ and $\gamma_u$) and exponentially decrease the number of samples per class. More experiment details are given in appendix section C.

In most experiments, we employ FixMatch to calculate the pseudo-label and make the prediction using the EMA version of the classifier following Sohn et al. (2020). We report the average test accuracy along with its variance, derived from three distinct random seeds.

## 5.1 MAIN RESULTS

| Algorithm | Method type | | | CIFAR10-LT $\gamma_l = \gamma_u = 100$ | | CIFAR100-LT $\gamma_l = \gamma_u = 10$ | | STL10-LT $\gamma_l = 20, \gamma_u$: *unknown* | |
|---|---|---|---|---|---|---|---|---|---|
| | LTL | PLR | THA | $n_1 = 500$ $m_1 = 4000$ | $n_1 = 1500$ $m_1 = 3000$ | $n_1 = 50$ $m_1 = 400$ | $n_1 = 150$ $m_1 = 300$ | $n_1 = 150$ | $n_1 = 450$ $M = 100,000$ |
| Supervised | | | | 47.3 ±0.95 | 61.9 ±0.41 | 29.6 ±0.57 | 46.9 ±0.22 | 39.4 ±1.40 | 51.7 ±2.21 |
| w/ LA Menon et al. (2020) | ✓ | | | 53.3 ±0.44 | 70.6 ±0.21 | 30.2 ±0.44 | 48.7 ±0.89 | 42.0 ±1.24 | 55.8 ±2.22 |
| FixMatch Sohn et al. (2020) | | | | 67.8 ±1.13 | 77.5 ±1.32 | 45.2 ±0.55 | 56.5 ±0.06 | 47.6 ±4.87 | 64.0 ±2.27 |
| w/ DARP Kim et al. (2020) | | ✓ | | 74.5 ±0.78 | 77.8 ±0.63 | 49.4 ±0.20 | 58.1 ±0.44 | 59.9 ±2.17 | 72.3 ±0.60 |
| w/ FlexMatch Zhang et al. (2021) | | | ✓ | 74.0 ±0.64 | 78.2 ±0.45 | 49.9 ±0.61 | 58.7 ±0.24 | 48.3 ±2.75 | 66.9 ±2.34 |
| w/ Adsh Guo & Li (2022) | | | ✓ | 73.0 ±3.46 | 77.2 ±1.01 | 49.6 ±0.64 | 58.9 ±0.71 | 60.0 ±1.75 | 71.4 ±1.37 |
| w/ FreeMatch Wang et al. (2022b) | | ✓ | ✓ | 73.8 ±0.87 | 77.7 ±0.23 | 49.8 ±1.02 | 59.1 ±0.59 | 63.5 ±2.61 | 73.9 ±0.48 |
| w/ SEVAL-PL | | ✓ | ✓ | **77.7** ±1.38 | **79.7** ±0.53 | **50.8** ±0.84 | **59.4** ±0.08 | **67.4** ±0.79 | **75.2** ±0.48 |
| w/ ABC Wei et al. (2021) | ✓ | | | 78.9 ±0.82 | 83.8 ±0.36 | 47.5 ±0.18 | 59.1 ±0.21 | 58.1 ±2.50 | 74.5 ±0.99 |
| w/ CReST+ Wei et al. (2021) | ✓ | ✓ | | 76.3 ±0.86 | 78.1 ±0.42 | 44.5 ±0.94 | 57.1 ±0.65 | 56.0 ±3.19 | 68.5 ±1.88 |
| w/ DASO Oh et al. (2022) | ✓ | ✓ | | 76.0 ±0.37 | 79.1 ±0.75 | 49.8 ±0.24 | 59.2 ±0.35 | 65.7 ±1.78 | 75.3 ±0.44 |
| w/ ACR Wei & Gan (2023) | ✓ | ✓ | ✓ | 80.2 ±0.78 | 83.8 ±0.13 | 50.6 ±0.13 | 60.7 ±0.23 | 65.6 ±0.11 | **76.3** ±0.57 |
| w/ SEVAL | ✓ | ✓ | ✓ | **82.8** ±0.56 | **85.3** ±0.25 | **51.4** ±0.95 | **60.8** ±0.28 | **67.4** ±0.69 | 75.7 ±0.36 |

Table 2: Accuracy on CIFAR10-LT, CIFAR100-LT and STL10-LT. We divide SSL algorithms into different groups including long-tailed learning (LTL), pseudo-label refinement (PLR) and threshold adjustment (THA). PLR and THA based methods only modify pseudo-label probability $q_i$ and threshold $\tau$, respectively. Best results within the same category are in **bold** for each configuration.

We compare SEVAL with different kinds of SSL algorithms and summarize the results of test accuracy in Table 2. In order to fairly compare the algorithm performance, in this table, we mark SSL algorithms based on the way they tackle the imbalance challenge. In particular, techniques such as DARP, which exclusively manipulate the probability of pseudo-labels $\pi$, are denoted as pseudo-label refinement (PLR). In contrast, approaches like FlexMatch, which solely alter the threshold $\tau$, are termed as threshold adjustment (THA). We denote other methods that apply regularization techniques to the model's cost function using labeled data as long-tailed learning (LTL). Besides the results from SEVAL, we also report results of SEVAL-PL, which forgoes any post-hoc adjustments on test samples. This ensures that its results are directly comparable with its counterparts.

As shown in Table 2, SEVAL-PL outperform other PLR and THA based methods such as DARP, FlexMatch and FreeMatch with a considerable margin. This indicates that SEVAL can provide better pseudo-label for the models by learning a better curriculum for $\pi$ and $\tau$.

When compared with other hybrid methods including ABC, CReST+, DASO, ACR, SEVAL demonstrates significant advantages in most scenarios. Relying solely on the strength of pseudo-labeling, SEVAL delivers highly competitive performance in the realm of imbalanced SSL. Importantly, given its straightforward framework, SEVAL can be integrated with other SSL concepts to enhance accuracy, a point we delve into later in the ablation study. We provide a summary of additional experimental results conducted under diverse realistic or extreme settings in appendix Section B.

### 5.1.1 THRESHOLD ADJUSTMENT

Quantity and quality are two crucial factors for pseudo-labels, as highlighted in Chen et al. (2023). Specifically, quantity denotes the count of accurately labeled samples generated by pseudo-label algorithms, whereas quality represents the ratio of accurately labeled samples after confidence-based thresholding. Having just high quantity or just high quality isn't enough for effective pseudo-labels. For instance, setting exceedingly high thresholds might lead to the selection of a limited number of accurately labeled samples (high quality). However, this is not always the ideal approach, and the opposite holds true for quantity.

In order to access the effectiveness of pseudo-label, we propose a metric called **correctness**, which is a combination of quantity and quality. Factoring in the potential imbalance of unlabeled data, we utilize a class frequency based weight term $\omega^{\mathcal{U}} = 1/\boldsymbol{m}$ to normalize this metric, yielding:

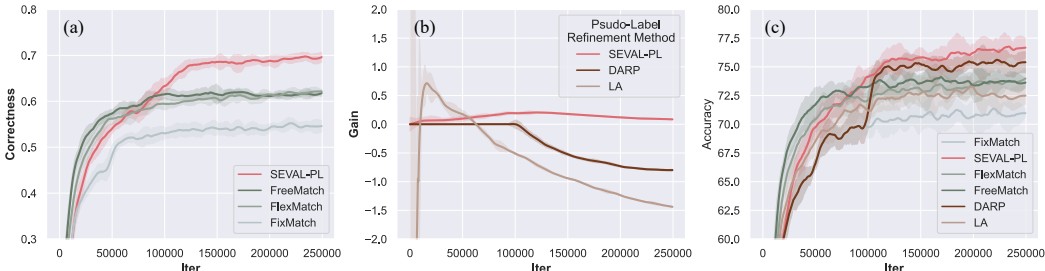

Figure 3: (a) The evolution of **Correctness** across training iterations. SEVAL can build better trade-off between quality and quantity. (b) The evolution of **Gain** across training iterations. SEVAL accumulates a higher accuracy advantage than its counterparts. (c) The evolution of test accuracy across training iterations. SEVAL-PL outperforms other pseudo-label refinement methods.

$$\textbf{Correctness} = \underbrace{\frac{\mathcal{C}}{\sum_{i=1}^{M} \omega_{y_i}^{\mathcal{U}}}}_{\text{Quantity}} \underbrace{\frac{\mathcal{C}}{\sum_{i=1}^{M} \omega_{y_i}^{\mathcal{U}} \mathbb{1}(\max_j(q_{ij}) \geq \tau_{y_i'})}}_{\text{Quality}}, \tag{5}$$

where, $\mathcal{C} = \sum_{i=1}^{M} \omega_{y_i}^{\mathcal{U}} \mathbb{1}(\hat{y}_i = y_i) \mathbb{1}(\max_j(q_{ij}) \geq \tau_{y'})$ is the relative number of correctly labelled samples. We show **correctness** of SEVAL with FixMatch, FlexMatch and FreeMatch in Figure 3(a). We observe that FlexMatch and FreeMatch can both improve **correctness**, while SEVAL can boost even more. We observe that the test accuracy follows a trend similar to **correctness**, as shown in Figure 3(c). This demonstrates that the thresholds set by SEVAL not only ensure a high quantity but also attain high accuracy for pseudo-labels, making them efficient in the model's learning process.

### 5.1.2 PSEUDO-LABEL REFINEMENT

Both sample-specific accuracy and class-specific accuracy are crucial measures to evaluate the quality of pseudo-labels. A low sample-specific accuracy can lead to noisier pseudo-labels, adversely affecting model performance. Meanwhile, a low class-specific accuracy often indicates a bias towards the dominant classes. Therefore, in order to comprehensively and quantitatively investigate the accuracy of pseudo-label refined by different approaches, here we define $G$ as the sum of accuracy gain and balanced accuracy gain of pseudo-label over training iterations. Specifically, given the pseudo-label $\hat{y}_i$ and predicted class $y_i'$ of unlabelled dataset $\mathcal{U}$, we calculate G as:

$$G = \underbrace{\frac{\sum_{i=1}^{M}[\mathbb{1}(\hat{y}_i' = y_i) - \mathbb{1}(\hat{y}_i = y_i)]}{M}}_{\text{Sample-Wise Accuracy Gain}} + \underbrace{\sum_{c=1}^{C}\sum_{i=1}^{M} \frac{\mathbb{1}(\hat{y}_i' = c)\mathbb{1}(\hat{y}_i' = y_i) - \mathbb{1}(\hat{y}_i = c)\mathbb{1}(\hat{y}_i = y_i)}{m_c C}}_{\text{Class-Wise Accuracy Gain}}. \tag{6}$$

To evaluate the cumulative impact of pseudo-labels, we calculate $G(\textbf{iter})$ as the accuracy gain at training iteration **iter** and monitor $\textbf{Gain} = \sum_{j=1}^{\textbf{iter}} G(j)/\textbf{iter}$ throughout the training iterations. The results of SEVAL along with DARP and adjusting pseudo-label logit $\hat{z}_c^{\mathcal{U}}$ with LA are summarized in Figure 3(b). We note that SEVAL consistently delivers a positive **Gain** throughout the training iterations. In contrast, DARP and LA tend to reduce the accuracy of pseudo-labels during the later stages of the training process.

After a warm-up period, DARP adjusts the distribution of pseudo-labels to match the inherent distribution of unlabeled data. However, it doesn't guarantee the accuracy of the pseudo-labels, thus not optimal. While LA can enhance class-wise accuracy, it isn't always the best fit for every stage of the model's learning. Consequently, noisy pseudo-labels from the majority class can impede the model's training. SEVAL learns a smooth curriculum of parameters for pseudo-label refinement from the data itself, therefore bringing more stable improvements. We can further validate the effectiveness of SEVAL from the test accuracy curves shown in Figure 3(c) where SEVAL-PL outperforms LA and DARP.

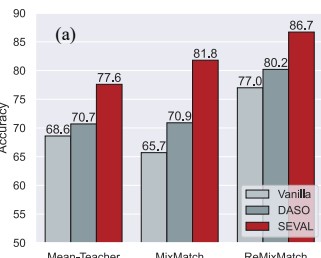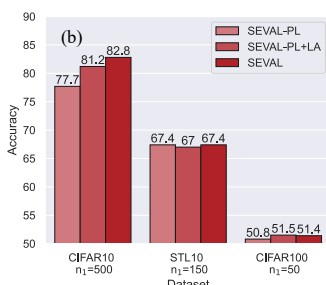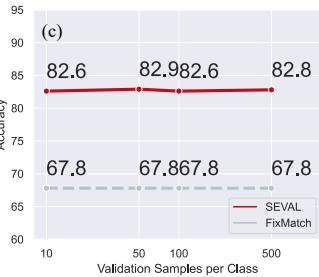

Figure 4: (a) Test accuracy when SEVAL is adapted to pseudo-label based SSL algorithms other than FixMatch under the setting of CIFAR-10 $n_1 = 1500$. SEVAL can readily improve the performance of other SSL algorisghm. (b) Test accuracy when SEVAL employs varied types of post-hoc adjustment parameters. The learned post-hoc parameters consistently enhance performance, particularly in CIFAR-10 experiments. (c) Test accuracy when SEVAL is optimized using different validation samples under the setting of CIFAR-10 $n_1 = 500$. SEVAL requires few validation samples to learn the optimal curriculum of parameters.

## 5.2 ABLATION STUDY

### 5.2.1 FLEXIBILITY AND COMPATIBILITY

We apply SEVAL to other pseudo-label based SSL algorithms including Mean-Teacher, MixMatch and ReMixMatch and report the results with the setting of CIFAR-100 $n_1 = 50$ in Figure 4(a). We find SEVAl can bring substantial improvements to these methods and is more effective than DASO. Of note the results of ReMixMatch w/SEVAL is higher than the results of FixMatch w/ SEVAL in Table 2 (86.7 vs 85.3). This may indicates that ReMixMatch is fit imbalanced SSL better. Due to its simplicity, SEVAL can be readily combined with other SSL algorithms that focus on LTL instead of PLR and THA. For example, SEVAL pairs effectively with the semantic alignment regularization introduced by DASO. By incorporating this loss into our FixMatch experiments with SEVAL, we were able to boost the test accuracy from 51.4 to 52.4 using the CIFAR-100 $n_1 = 50$ configuration.

We compared with the post-hoc adjustment process with LA in Figure 4(b). We find that the post-hoc parameters can improve the model performance significantly in the setting of CIFAR-10. In other cases, our post-hoc adjustment doesn't lead to a decrease in prediction accuracy. However, LA sometimes does, as seen in the case of STL-10. This could be due to the complexity of the confusion matrix in those instances, where the class bias is not adequately addressed by simple offsets.

### 5.2.2 DATA-EFFICIENCY

Here we explore if SEVAL requires a substantial number of validation samples for curriculum learning. To do so, we keep the training dataset the same and optimize SEVAL parameters using balanced validation dataset with varied numbers of labelled samples using the CIFAR-10 $n_1 = 500$ configuration, as shown in Figure 4(c). We find that SEVAL consistently identifies similar $\pi$ and $\tau$. When we train the model using these curricula, there aren't significant differences even when the validation samples per class ranges from 10 to 500. This suggests that SEVAL is both data-efficient and resilient. We conduct stress tests on SEVAL and observe its effectiveness, even with only 40 labelled samples in total, as detailed in the appendix Section B.3.

## 6 CONCLUSION AND FUTURE WORK

In this study, we present SEVAL and highlight its benefits in imbalanced SSL across a wide range of application scenarios. SEVAL sheds new light on pseudo-label generalization, a foundation for many leading SSL algorithms. SEVAL is both straightforward and potent, requiring no extra computation once the curriculum is acquired. As such, it can effortlessly be integrated into other SSL algorithms and paired with LTL methods to address class imbalance. Moreover, we believe that the concept of optimizing parameters or accessing unbiased learning status using a partition of labelled training dataset could spark further innovations in long-tailed recognition and SSL. We feel the specific interplay between label refinement and threshold adjustment remains an intriguing question for subsequent research.

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

## A  ALGORITHM

---

**Algorithm 1** SEVAL parameter estimation process, $\boldsymbol{\pi}^*, \boldsymbol{\tau}^* \leftarrow \text{ESTIM}\left(\mathcal{V}, \{\boldsymbol{z}_i^{\mathcal{V}}\}_{i=1}^K\right)$

---

**Require:**

$\mathcal{V} = \{(\boldsymbol{x}_i, y_i)\}_{i=1}^K$: validation data, $\{\boldsymbol{z}_i^{\mathcal{V}}\}_{i=1}^K$: network prediction of $\mathcal{V}$.

$C$: Number of classes, $t$: Requested per class accuracy of the pseudo-label, $\boldsymbol{k}$: Number of sample per class for $\mathcal{V}$.

1: $\boldsymbol{\pi}^* = \arg\min_{\boldsymbol{\pi}} \frac{1}{K} \sum_{i=1}^K \mathcal{H}(y_i, \sigma(\boldsymbol{z}_i^{\mathcal{V}} - \log \boldsymbol{\pi}))$

    ▷ *In practice, the parameter estimation process is achieved by bound-constrained solvers.*

2: $\boldsymbol{\omega}^{\mathcal{V}} = 1/\boldsymbol{k}$   ▷ *The minority class is assigned higher weights to prioritize class-specific accuracy.*

3: **for** c in C **do**

4:    Calculate class-wise accuracy $\alpha_c = \frac{1}{K_c} \sum_{i=1}^K \omega_{y_i}^{\mathcal{V}} \mathbb{1}_{ic} \mathbb{1}(y_i = c)$

       ▷ *For each class c, $\mathbb{1}_{ic} = \mathbb{1}(\arg\max_j(p_{ij}^{\mathcal{V}}) = c)$ and $K_c = \sum_{i=1}^K \omega_{y_i}^{\mathcal{V}} \mathbb{1}_{ic}$*

5:    **if** $\alpha_c > t$ **then**

6:        $\tau_c^* = \arg\min_{\tau_c} \left| \frac{1}{s_c} \sum_{i=1}^K \omega_{y_i}^{\mathcal{V}} \mathbb{1}_{ic} \mathbb{1}(y_i = c) \mathbb{1}(\max_j(p_{ij}^{\mathcal{V}}) > \tau_c) - t \right|$

       ▷ *$s_c = \sum_{i=1}^K \omega_{y_i}^{\mathcal{V}} \mathbb{1}_{ic} \mathbb{1}(\max_j(p_{ij}^{\mathcal{V}}) > \tau_c)$ is the relative number of samples predicted as class c with confidence larger than $\tau_c$*

7:    **else**

8:        $\tau_c^* = 0$   ▷ *The quality of the pseudo-labels is satisfactory, and we make use of all of them.*

9:    **end if**

10: **end for**

---

**Algorithm 2** Imbalanced semi-supervised learning with SEVAL.

---

**Require:**

$\mathcal{X} = \{(\boldsymbol{x}_i, y_i)\}_{i=1}^N$: labelled training data, $\mathcal{U} = \{\boldsymbol{u}_i\}_{i=1}^M$: unlabelled training data, $f(\cdot)$: network for classification.

$T$: Total training iterations, $C$: Number of classes, $L$: length of the curriculum, $\rho_\pi, \rho_\tau$: Momentum decay ratio of offsets and thresholds.

1: Initialize the SEVAL parameters as $l = 1$, $\boldsymbol{\pi}^{(l)} = \underbrace{[1, 1, \ldots, 1]}_{C}$ and $\boldsymbol{\tau}^{(l)} = \underbrace{[0.95, 0.95, \ldots, 0.95]}_{C}$.

    ▷ *Estimate a curriculum of the SEVAL parameters based on a partition of the training dataset.*

2: Randomly partition $\mathcal{X}$ into two subsets, $\mathcal{X}' = \{(\boldsymbol{x}_i, y_i)\}_{i=1}^K$ and $\mathcal{V}' = \{(\boldsymbol{x}_i, y_i)\}_{i=1}^K$, each containing an equal number of data points.

3: **for iter** in $[1, \ldots, T]$ **do**

4:    Calculate the pseudo-label logit for unlabelled data $\mathcal{U}$ and obtain $\{\hat{\boldsymbol{z}}_i^{\mathcal{U}}\}_{i=1}^M$. ▷ *Note: FixMatch achieves this by utilizing two augmented versions of the unlabelled data.*

5:    Calculate the pseudo-label probability $\boldsymbol{q}_i = \sigma(\hat{\boldsymbol{z}}_i^{\mathcal{U}} - \log \boldsymbol{\pi}^{(l)})$.

6:    For pseudo-label $\hat{y}_i = \arg\max_j q_{ij}$ and predicted class $y_i' = \arg\max_j p_{ij}^{\mathcal{U}}$, calculate the unlabelled loss $\mathcal{L}_{\mathrm{u}} = \frac{1}{M} \sum_{i=1}^M \mathbb{1}(\max_j(q_{ij}) \geq \tau_{y_i'}^{(l)}) \mathcal{H}(\hat{y}_i, \boldsymbol{p}_i^{\mathcal{U}})$.

7:    Update the network $f$ with labelled loss $\mathcal{L}_{\mathrm{cls}}$ calculated using $\mathcal{X}'$ and $\mathcal{L}_u$ via SGD optimizer.

8:    **if iter** $\% (T/L) = 0$ **then**

9:        $l = \textbf{iter} L/T$

10:       Calculate the prediction on $\mathcal{V}'$ using EMA model and obtain $\{\boldsymbol{z}_i^{\mathcal{V}}\}_{i=1}^K$.

11:       $\boldsymbol{\pi}^{(l)*}, \boldsymbol{\tau}^{(l)*} = \text{ESTIM}(\mathcal{V}', \{\boldsymbol{z}_i^{\mathcal{V}}\}_{i=1}^K)$

12:       $\boldsymbol{\pi}^{(l)} = \rho_\pi \boldsymbol{\pi}^{(l-1)} + (1 - \rho_\pi) \boldsymbol{\pi}^{(l)*}$, $\boldsymbol{\tau}^{(l)} = \rho_\tau \boldsymbol{\tau}^{(l-1)} + (1 - \rho_\tau) \boldsymbol{\tau}^{(l)*}$

13:    **end if**

14: **end for**

    ▷ *Standard SSL process.*

15: **for iter** in $[1, \ldots, T]$ **do**

16:    $l = \lceil \textbf{iter} L/T \rceil$

17:    Calculate the pseudo-label logit for unlabelled data $\mathcal{U}$ and obtain $\{\hat{\boldsymbol{z}}_i^{\mathcal{U}}\}_{i=1}^M$.

18:    Calculate the pseudo-label probability $\boldsymbol{q}_i = \sigma(\hat{\boldsymbol{z}}_i^{\mathcal{U}} - \log \boldsymbol{\pi}^{(l)})$.

19:    Calculate the unlabelled loss $\mathcal{L}_{\mathrm{u}} = \frac{1}{M} \sum_{i=1}^M \mathbb{1}(\max_j(q_{ij}) \geq \tau_{y_i'}^{(l)}) \mathcal{H}(\hat{y}_i, \boldsymbol{p}_i^{\mathcal{U}})$.

20:    Update the network $f$ with labelled loss $\mathcal{L}_{\mathrm{cls}}$ calculated using $\mathcal{X}$ and $\mathcal{L}_u$ via SGD optimizer.

21: **end for**

    ▷ *Post-hoc processing with final learned parameters.*

22: Given a test sample $\boldsymbol{x}_i$, the logit is adjusted from $\boldsymbol{z}_i$ to $\boldsymbol{z}_i - \log \boldsymbol{\pi}^{(L)*}$.

# B ADDITIONAL EXPERIMENTS

In this section, we present additional experimental results conducted under various settings to assess the generalizability of SEVAL.

## B.1 VARIED IMBALANCED RATIOS

Similar to results in Table 2, we evaluate SEVAL on CIFAR10-LT with different imbalanced ratios. We find SEVAL consistently outperform its counterparts with different $\gamma_l$. Since SEVAL doesn't make any assumptions about the distribution of unlabeled data, it can be robustly implemented in this context when $\gamma_l \neq \gamma_u$. We find SEVAL can outperform its counterparts even more under such settings.

|  | Method type | | | CIFAR10-LT | | | | | |
|  |  | | | $\gamma_l = 100, \gamma_u = 1$ | | $\gamma_l = 100, \gamma_u = 1/100$ | | $\gamma_l = \gamma_u = 150$ | |
| Algorithm | LTL | PLR | THA | $n_1 = 500$ $m_1 = 4000$ | $n_1 = 1500$ $m_1 = 3000$ | $n_1 = 500$ $m_1 = 4000$ | $n_1 = 1500$ $m_1 = 3000$ | $n_1 = 500$ $m_1 = 4000$ | $n_1 = 1500$ $m_1 = 3000$ |
|---|---|---|---|---|---|---|---|---|---|
| FixMatch Sohn et al. (2020) | | | | 73.0 ±3.81 | 81.5 ±1.15 | 62.5 ±0.94 | 71.8 ±1.70 | 62.9 ±0.36 | 72.4 ±1.03 |
| w/ DARP Kim et al. (2020) | | ✓ | | 82.5 ±0.75 | 84.6 ±0.34 | 70.1 ±0.22 | 80.0 ±0.93 | 67.2 ±0.32 | 73.6 ±0.73 |
| w/ SEVAL-PL | | ✓ | ✓ | **89.4** ±0.53 | **89.2** ±0.02 | **77.7** ±0.91 | **80.9** ±0.66 | **71.9** ±1.10 | **74.7** ±0.63 |
| w/ CReST+ Wei et al. (2021) | ✓ | ✓ | | 82.2 ±1.53 | 86.4 ±0.42 | 62.9 ±1.39 | 72.9 ±2.00 | 67.5 ±0.45 | 73.7 ±0.34 |
| w/ DASO Oh et al. (2022) | ✓ | ✓ | | 86.6 ±0.84 | 88.8 ±0.59 | 71.0 ±0.95 | 80.3 ±0.65 | 70.1 ±1.81 | 75.1 ±0.77 |
| w/ SEVAL | ✓ | ✓ | ✓ | **90.3** ±0.61 | **90.6** ±0.47 | **79.2** ±0.83 | **82.9** ±1.78 | **79.8** ±0.42 | **83.3** ±0.40 |

Table 3: Accuracy on CIFAR10-LT with different imbalanced ratios. Best results within the same category are in **bold** for each configuration.

## B.2 RESULTS ON SEMI-AVES

We further apply SEVAL to the realistic imbalanced SSL dataset, Semi-Aves Su & Maji (2021), which captures a situation where a portion of the unlabelled data originates from previously unseen classes. This dataset, contained 200 classes with different with long-tailed distribution. In addition to labelled data, Semi-Aves also contains imbalanced unlabelled data $\mathcal{U}_{in}$ and unlabelled open-set data $\mathcal{U}_{out}$ from other 800 classes. Following previous works Su et al. (2021); Oh et al. (2022), we conduct experiments using $\mathcal{U}_{in}$ or a combination of $\mathcal{U}_{in}$ and $\mathcal{U}_{out}$.

We summarize the results in Table 4. This dataset poses a challenge due to the limited number of samples in the tail class, with only around 15 samples per class. It has been observed that SEVAL performs effectively in such a demanding scenario.

|  | Method type | | | Semi-Aves | |
| Algorithm | LTL | PLR | THA | $\mathcal{U} = \mathcal{U}_{in}$ | $\mathcal{U} = \mathcal{U}_{in} + \mathcal{U}_{out}$ |
|---|---|---|---|---|---|
| FixMatch Sohn et al. (2020) | | | | 59.9 ±0.08 | 52.6 ±0.14 |
| w/ DARP Kim et al. (2020) | | ✓ | | 60.3 ±0.24 | 54.7 ±0.06 |
| w/ SEVAL-PL | | ✓ | ✓ | **60.6** ±0.18 | **56.4** ±0.10 |
| w/ CReST+ Wei et al. (2021) | ✓ | ✓ | | 60.0 ±0.03 | 54.3 ±0.59 |
| w/ DASO Oh et al. (2022) | ✓ | ✓ | | 59.3 ±0.28 | 56.6 ±0.32 |
| w/ SEVAL | ✓ | ✓ | ✓ | **60.7** ±0.17 | **56.7** ±0.15 |

Table 4: Accuracy on Semi-Aves. Best results within the same category are in **bold** for each configuration.

### B.3 Low labelled data scheme

SEVAL acquires a curriculum of parameters by partitioning the training dataset. An inherent question arises: can SEVAL still be effective when there are extremely limited labeled samples? In this context, we subject SEVAL to a stress-test by training it with a very small amount of labeled training data.

In the first experimental configuration, we maintain the imbalance ratio while diminishing the count of labeled samples ($n_1 = 200$). In this extreme scenario, only two samples are labeled for the tail class. In the second configuration, we employ a balanced labeled training dataset, but with a total of 100 and 40 samples for training.

We summarize the results in Table 5. We find SEVAL work well in both settings. This indicates SEVAL can be a safe choice even when very few labelled dataset.

| | Method type | | | CIFAR10-LT | | |
| | | | | $\gamma_l = \gamma_u = 100$ | $\gamma_l = 1, \gamma_u = 100$ | |
| Algorithm | LTL | PLR | THA | $n_1 = 200$ $m_1 = 4000$ | $n_1 = 10$ $m_1 = 4000$ | $n_1 = 4$ $m_1 = 4000$ |
|---|---|---|---|---|---|---|
| FixMatch Sohn et al. (2020) | | | | 64.3 ±0.83 | 65.3 ±0.80 | 44.7 ±3.33 |
| w/ FreeMatch Wang et al. (2022b) | | ✓ | ✓ | 67.4 ±1.09 | 58.4 ±0.76 | 50.7 ±1.95 |
| w/ SEVAL-PL | | ✓ | ✓ | **69.3** ±0.66 | **68.3** ±0.56 | **51.5** ±1.51 |
| w/ DASO Oh et al. (2022) | ✓ | ✓ | | 67.2 ±1.25 | 61.2 ±0.96 | 48.6 ±2.81 |
| w/ SEVAL | ✓ | ✓ | ✓ | **71.2** ±0.80 | **68.9** ±0.25 | **52.7** ±1.83 |

Table 5: Accuracy on CIFAR10-LT under the setting of extremely few labeled samples. Best results within the same category are in **bold** for each configuration.

### B.4 Ablation study

To closely examine the distinct contributions of $\pi$ and $\tau$, we carry out an ablation study where SEVAL optimizes just one of them, respectively termed SEVAL-PLR and SEVAL-THA. As summarized in Table 6, SEVAL-PLR and SEVAL-THA can still outperform their counterparts, DARP and FlexMatch, respectively. When tuning both parameters, SEVAL-PL can achieve the best results.

| | Method type | | | CIFAR10-LT | CIFAR100-LT |
| | | | | $n_1 = 500, m_1 = 4000$ | $n_1 = 150, m_1 = 300$ |
| Algorithm | LTL | PLR | THA | $\gamma_l = \gamma_u = 100$ | $\gamma_l = \gamma_u = 10$ |
|---|---|---|---|---|---|
| FixMatch Sohn et al. (2020) | | | | 67.8 ±1.13 | 56.5 ±0.06 |
| w/ DARP Kim et al. (2020) | | ✓ | | 74.5 ±0.78 | 58.1 ±0.44 |
| w/ SEVAL-PLR | | ✓ | | 76.7 ±0.82 | 59.3 ±0.30 |
| w/ FlexMatch Zhang et al. (2021) | | | ✓ | 74.0 ±0.64 | 58.7 ±0.24 |
| w/ SEVAL-THA | | | ✓ | 77.0 ±0.93 | 59.1 ±0.18 |
| w/ SEVAL-PL | | ✓ | ✓ | **77.7** ±1.38 | **59.4** ±0.08 |

Table 6: Comparison of SEVAL when only optimizing $\pi$ (SEVAL-PLR) or only optimizing $\tau$ (SEVAL-THA). SEVAL outperforms counterparts with identical parameter settings under different imbalanced SSL scenarios. SEVAL-PL, with its sequential optimization of both $\pi$ and $\tau$, yields further improvements in accuracy.

### B.5 INTEGRATION WITH OTHER SSL FRAMEWORKS

As an extension to results in Figure 4, we summarize the results when introducing SEVAL into other SSL frameworks in Table 7.

We summarize the implementation details of those methods in Section C.

| Algorithm | CIFAR10-LT $n_1 = 1500, m_1 = 3000$ $\gamma_l = \gamma_u = 100$ | CIFAR100-LT $n_1 = 150, m_1 = 300$ $\gamma_l = \gamma_u = 10$ |
|---|---|---|
| Mean Teacher Tarvainen & Valpola (2017) | 68.6 ±0.88 | 52.1 ±0.09 |
| w/ DASO Oh et al. (2022) | 70.7 ±0.59 | 52.5 ±0.37 |
| w/ SEVAL | **77.6** ±0.63 | **53.8** ±0.24 |
| MixMatch Berthelot et al. (2019b) | 65.7 ±0.23 | 54.2 ±0.47 |
| w/ DASO Oh et al. (2022) | 70.9 ±1.91 | 55.6 ±0.49 |
| w/ SEVAL | **81.8** ±0.82 | **57.8** ±0.26 |
| ReMixMatch Berthelot et al. (2019a) | 77.0 ±0.55 | 61.5 ±0.57 |
| w/ DASO Oh et al. (2022) | 80.2 ±0.68 | 62.1 ±0.69 |
| w/ SEVAL | **86.7** ±0.71 | **63.1** ±0.38 |

Table 7: Accuracy on CIFAR10-LT based on SSL methods other than FixMatch. Best results within the same category are in **bold** for each configuration.

### B.6 SENSITIVITY ANALYSIS

We perform experiments with SEVAL, varying the core hyperparameters, and present the results in Table 8. Our findings indicate that SEVAL exhibits robustness, showing insensitivity to hyperparameter variations within a reasonable range.

| Hyper-parameter | CIFAR10-LT, $\gamma_l = 100$ $n_1 = 500, m_1 = 4000$ |
|---|---|
| $t = 0.6$ | 82.5 ±0.45 |
| $t = 0.7$ | 82.2 ±0.11 |
| $t = 0.75$ (reported) | **82.8** ±0.56 |
| $\rho_\pi = 0.995$ | 81.4 ±0.36 |
| $\rho_\pi = 0.999$ (reported) | **82.8** ±0.56 |
| $\rho_\pi = 0.9995$ | 82.5 ±0.35 |
| $\rho_\tau = 0.995$ | 81.5 ±0.38 |
| $\rho_\tau = 0.999$ (reported) | 82.8 ±0.56 |
| $\rho_\tau = 0.9995$ | **82.9** ±0.09 |

Table 8: Sensitivity analysis of hyper-parameters $t$, $\rho_\pi$ and $\rho_\tau$. Best results are in **bold** for each configuration.

## C IMPLEMENTATION DETAILS

### C.1 LEARNING WITH IMBALANCED VALIDATION DATA

As the labelled training dataset $\mathcal{X}$ is imbalanced, in practice, it is hard to obtain a balanced split $\mathcal{V}$ to learn a curriculum of threshold $\tau$. However, when we optimize $\tau$ using an imbalanced validation $\mathcal{V}$ following Eq. 4, the optimized results would be biased. More precisely, the majority class consistently exhibits high precision, leading to a lower threshold, while the opposite holds true for the minority class.

Therefore, we utilize the class frequency of the labelled validation data $k$ to normalize the cost function. Specifically, we calculate the class weight as $\omega^{\mathcal{V}} = 1/k$. Then we replace all the $\mathbb{1}_{ic}$ with $\omega_{y_i}^{\mathcal{V}} \mathbb{1}_{ic}$ in Eq. 4, obtaining:

$$\tau_c^* = \begin{cases} \arg\min_{\tau_c} \left| \frac{1}{s_c} \sum_{i=1}^K \omega_{y_i}^{\mathcal{V}} \mathbb{1}_{ic} \mathbb{1}(y_i = c) \mathbb{1}(\max_j(p_{ij}^{\mathcal{V}}) > \tau_c) - t \right| & \text{if} \quad t < \alpha_c \\ 0 & \text{otherwise} \end{cases}, \quad (7)$$

where $s_c = \sum_{i=1}^K \omega_{y_i}^{\mathcal{V}} \mathbb{1}_{ic} \mathbb{1}(\max_j(p_{ij}^{\mathcal{V}}) > \tau_c)$ is the relative number of samples predicted as class $c$ with confidence larger than $\tau_c$, where $\alpha_c = \frac{1}{K_c} \sum_{i=1}^K \omega_{y_i}^{\mathcal{V}} \mathbb{1}_{ic} \mathbb{1}(y_i = c)$ is the average balanced accuracy of all the samples predicted as class $c$ and $K_c = \sum_{i=1}^K \omega_{y_i}^{\mathcal{V}} \mathbb{1}_{ic}$ is the relative number of samples predicted as $c$.

This modification can normalize the number of samples within the cost function. Consequently, we can directly learn the thresholds $\boldsymbol{\tau}$ using imbalanced validation data.

## C.2 LEARNING THRESHOLDS WITHIN GROUPS

When we learn $\boldsymbol{\tau}$ based on the validation data $\mathcal{V}$, the optimization process could be unstable as sometimes we have very few samples per class (e.g. less than 10 samples). In this case, even if we can re-weight the validation samples based on their class prior $\boldsymbol{k}$, it is hard to have enough samples to obtain stable $\boldsymbol{\tau}$ curriculum for the minority classes, especially when $\min_c k_c < 10$. Assuming equal class priors should result in similar thresholds, we propose to optimize thresholds within groups, pinpointing the ideal ones that fulfill the accuracy requirement for every classes within the group.

We assume the samples of different classes $k_c$ are arranged in descending order. In other words, $k_1$ is the maximum, and $k_C$ is the minimum. Instead of optimizing $\tau_c$ for an individual class $c$, we optimize for groups such that the learned $\tau_b$ can satisfy the accuracy requirements for $B$ classes. Specifically, the optimal $\tilde{\boldsymbol{\tau}} \in \mathbb{R}^{C/B}$ is determined as:

$$\tilde{\tau}_b^* = \begin{cases} \sum_{c=bB+1}^{bB+B} \arg\min_{\tilde{\tau}_b} \left| \frac{1}{\tilde{s}_b} \sum_{i=1}^K \mathbb{1}_{ic} \mathbb{1}(y_i = c) \mathbb{1}(\max_j(p_{ij}^{\mathcal{V}}) > \tilde{\tau}_b) - t \right| & \text{if} \quad t < \tilde{\alpha}_b \\ 0 & \text{otherwise} \end{cases}, \quad (8)$$

where $\tilde{s}_b = \sum_{c=bB+1}^{bB+B} \sum_{i=1}^K \mathbb{1}_{ic} \mathbb{1}(\max_j(p_{ij}^{\mathcal{V}}) > \tilde{\tau}_b)$ is the number of samples that are chosen in this group based on the threshold $\tilde{\tau}_b$ and $\tilde{\alpha}_b = \frac{1}{\sum_{c=bB+1}^{bB+B} K_c} \sum_{c=bB+1}^{bB+B} \sum_{i=1}^K \mathbb{1}_{ic} \mathbb{1}(y_i = c)$ is the average accuracy of all the samples predicted as class in this group. If we set $B = 1$, Eq. 8 becomes equivalent to Eq. 4.

Furthermore, in practice, we find that in the setting of imbalanced SSL, sometimes the minority classes very few samples and the thresholds cannot be optimized correctly based on Eq. 8. In this case, we also set the learned $\tilde{\tau}_b^*$ to be 0, in order to leverage more data from the minority classes. Formally, we denote $\tilde{K}_b = \sum_{c=bB+1}^{bB+B} \sum_{i=1}^K \omega^{\mathcal{V}} \mathbb{1}_{ic}$ as the relative number of predicted samples within group $b$. When $\tilde{K}_b < \sum_{i=1}^K \frac{B\omega^{\mathcal{V}}}{e_1 C}$ or $\sum_{i=1}^K \mathbb{1}(y_i = c) < e_2$, where $e_1$ and $e_2$ are hyper-parameters that we both set to 10 for all experiments, we also have $\tilde{\tau}_b^* = 0$ and keep their corresponding $\pi_c$ within group $b$ as low as $\pi_c = \min_j(\pi_j)$. This implies:

- In instances where the models exhibit a pronounced bias, limiting their capability to detect over 10% of the samples within a particular group, we adjust the associated thresholds and consequently increase our sample selection.

- When a group comprises fewer than 10 samples, the feasibility of optimizing thresholds based on proportion diminishes, necessitating an enhanced sample selection.

## C.3 BENCHMARKS

We conduct experiments following Oh et al. (2022) for experiments of CIFAR10-LT, CIFAR100-LT and STL10-LT. We take some baseline results from the DASO paper Oh et al. (2022) to Table 2, Table 3 and Table 7. including the results of supervised baselines, DARP, CReST+, ABC and DASO.

As DASO Oh et al. (2022) does not supply the code for the Semi-Aves experiments, we conduct all the experiments for this setting ourselves. We train ResNet-50 He et al. (2016) which is pretrained on

ImageNet Deng et al. (2009) for the task of Semi-Aves following Su & Maji (2021). In accordance with Oh et al. (2022), we merge the training and validation datasets provided by the challenge, yielding a total of 5,959 samples for training which come from 200 classes. We conduct experiments utilizing 26,640 unlabeled samples which share the same label space with $\mathcal{X}$ in the $\mathcal{U} = \mathcal{U}_{in}$ setting, and 148,848 unlabeled samples of which 122,208 are from open-set classes in the $\mathcal{U} = \mathcal{U}_{in} + \mathcal{U}_{out}$ setting. For experiments on Semi-Aves, we set the base learning rate as 0.005. We train the network for 45,000 iterations. The learning rate is linear warmed up during the first 25,00 iterations, and degrade after 15,000 and 30,000, with a factor of 10. We choose training batch size as 32. The images are firstly cropped to $256 \times 256$. During training, the images are then randomly cropped to $224 \times 224$. At inference time, the images are cropped in the center with size $224 \times 224$.

## C.4 HYPER-PARAMETERS

Here we summarize all the hyper-parameters we choose in this experiments to ease reproducibility.

| Hyper-parameter | CIFAR10-LT, $\gamma_l = 100$ $n_1 = 500$   $n1 = 1500$ $m_1 = 4000$   $m_1 = 3000$ | | CIFAR100-LT, $\gamma_l = 10$ $n_1 = 50$   $n_1 = 150$ $m_1 = 400$   $m_1 = 300$ | | STL10-LT, $\gamma_l = 20$ $n_1 = 150$   $n_1 = 450$ $M = 100,000$ | | Semi-Aves $\mathcal{U} = \mathcal{U}_{in}$   $\mathcal{U} = \mathcal{U}_{in} + \mathcal{U}_{out}$ | |
|---|---|---|---|---|---|---|---|---|
| $T$ | 250,000 | | 250,000 | | 250,000 | | 45,000 | |
| $t$ | 0.75 | | 0.5 | 0.65 | 0.7 | 0.6 | 0.9 | 0.99 |
| $t^{\mathcal{V}}$ | 0.9 | | 0.65 | 0.7 | 0.95 | 0.85 | —— | |
| $L$ | 500 | | 100 | | 500 | | 90 | |
| $C$ | 10 | | 100 | | 10 | | 200 | |
| $B$ | 2 | | 25 | 10 | 2 | 1 | 10 | |
| $\rho_\pi$ | 0.999 | | 0.95 | 0.9 | 0.995 | | 0.99 | 0.9 |
| $\rho_\tau$ | 0.999 | | 0.95 | 0.9 | 0.9995 | 0.999 | 0.99 | 0.9 |

Table 9: Experiment-specific hyper-parameters. $t^{\mathcal{V}}$ is the required accuracy if we directly optimize $\tau$ along the training process using a separate validation dataset.

## C.5 SEVAL WITH OTHER SSL ALGORITHMS

Here, we provide implementation details of how SEVAL can be integrated into other pseudo-labeling based SSL algorithms. Specifically, we apply SEVAL to Mean Teacher Tarvainen & Valpola (2017), MixMatch Berthelot et al. (2019b) and ReMixMatch Berthelot et al. (2019a). These algorithms produce pseudo-label $\hat{y}_i$ based on its corresponding pseudo-label probability $\boldsymbol{q}_i$ and logit $\hat{z}_i^{\mathcal{U}}$ in different ways. SEVAL can be easily adapted by refining $\boldsymbol{q}_i$ using the learned offset $\boldsymbol{\pi}$.

It should be noted that these SSL algorithms do not include the process of filtering out pseudo-labels with low confidence. Therefore, for simplicity and fair comparison, we do not include the threshold adjustment into these methods. We expect that SEVAL can enhance performance through threshold adjustment and plan to explore this further in the future.

### C.5.1 MEAN TEACHER

Mean Teacher generates pseudo-label logit $\hat{z}_i^{\mathcal{U}}$ based on a EMA version of the prediction models. SEVAL calculates the pseudo-label probability as $\boldsymbol{q}_i = \sigma(\hat{z}_i^{\mathcal{U}} - \log \boldsymbol{\pi}^*)$, which is expected to have less bias towards the majority class.

### C.5.2 MIXMATCH

MixMatch calculates $\hat{y}_i$ based on multiple transformed version of an unlabelled sample $\boldsymbol{u}_i$. SEVAL adjusts each one of them with $\boldsymbol{\pi}$, separately.

### C.5.3 REMIXMATCH

ReMixMatch proposes to refine pseudo-label probability $\boldsymbol{q}_i$ with distribution alignment to match the marginal distributions. SEVAL adjusts the the probability using $\boldsymbol{q}_i = \sigma(\hat{z}_i^{\mathcal{U}} - \log \boldsymbol{\pi}^*)$ before ReMixMatch's process including distribution alignment and temperature sharpening.

# D   ANALYSIS OF LEARNED THRESHOLDS

We try to determine the effectiveness of thresholds by looking into the precision of different classes, which should serve as approximate indicators of suitable thresholds. We show an example when optimized thresholds and learning status of FixMatch on CIFAR-10 $n_1 = 500$ in Figure 5, SEVAL learned $\tau_c$ is low for the class that have high precision. In contrast, MCP $P'_c$, does not show clear correction with precision. Specifically, as highlighted with the red circle, $P_c$ remains high for classes that exhibit high precision. Consequently, confidence-based threshold methods such as FlexMatch will tune the threshold to be high for classes with large $P_c$, inadequately addressing *Case 3* and *Case 4* as elaborated upon in the method section.

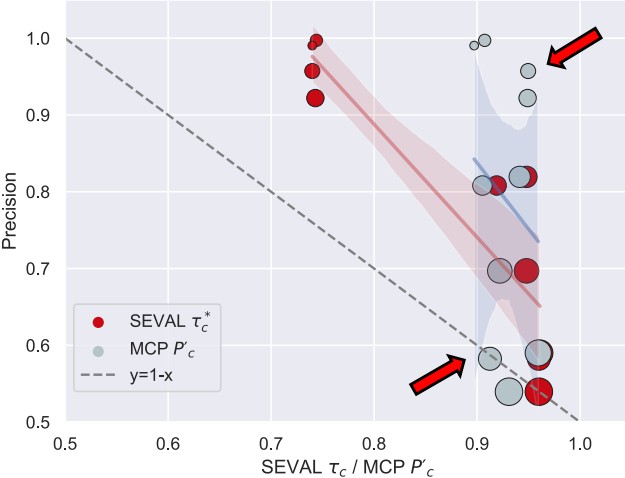

Figure 5: The correlation of SEVAL learned $\tau_c$ and MCP $P'_c$ between test precision of FixMatch on CIFAR10. Each point represents a class c and the size of the points indicate the number of samples in the labelled training dataset $n_c$. Note the $P'_c$ is the basis of current dynamic threshold method to derive thresholds. However, as highlighted by red arrows, $P'_c$ does not correlated with precision thus $P_c$ based on methods will fail *Case 3* and *Case 4* in Figure 2.

# E   FINE-GRAINED EVALUATION

Here we report class-wise performance of SEVAL and its counterparts in Figure 6. SEVAL can make neural networks more sensitive to minority classes.

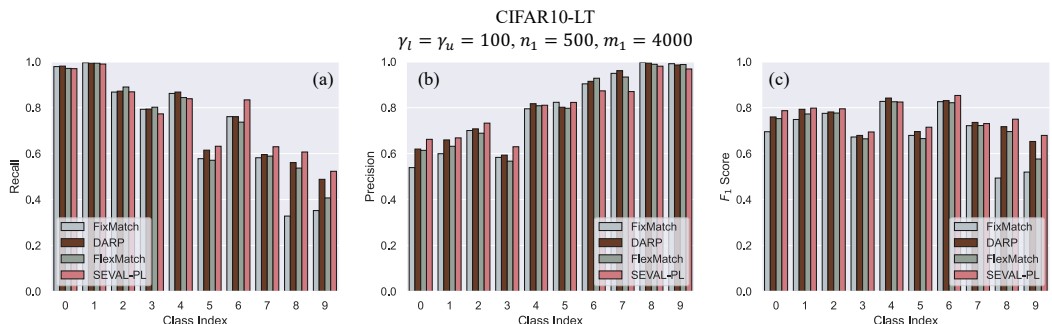

Figure 6: Class-wise performance for different SSL methods. Class indexes are arranged in descending order according to their class frequencies. When compared with alternative methods, SEVAL achieves overall better performance with higher recall on minority classes and higher precision on majority classes.

