# OpenReview forum: "Learning Label Refinement and Thresholds for Imbalanced Semi-Supervised Learning"
_ICLR.cc/2024/Conference — Submitted to ICLR 2024_

### Official Review · Reviewer_PXgw · 2023-10-15

**Soundness:** 1 poor
**Presentation:** 3 good
**Contribution:** 1 poor
**Rating:** 3
**Confidence:** 5

**Summary:**

The authors propose an algorithm termed SEVAL for imbalanced SSL which trains the algorithm on class imbalanced dataset consists of both labeled and unlabeled sets. To relieve classifier bias and thereby enhance the quality of pseudo-labels, SEVAL develop a curriculum for adjusting logits, and they also establish a curriculum for class-specific thresholds. Experimental results on CIFAR-10, 100 and STL-10 verify effectiveness of the proposed algorithm.

**Strengths:**

Code is submitted.

Many studies are reviewed in Section 2.

The paper is easy to read.

**Weaknesses:**

There is no theoretical ground for the proposed algorithm.

For CIFAR-10, the experimental settings are not enough. For example, other LTSSL (long tailed semi supervised learning) studies such as  DARP, DASO, SAW conducted experiments on CIFAR-10 with changing both imbalanced ratio of labeled and unlabeled sets.

The proposed algorithm appears to require extensive hyperparameter tuning.

The proposed algorithm and baseline algorithms are combined with only FixMatch, without being combined with ReMixMatch (other LTSSL studies such as DARP, ABC, SAW and CoSSL conducted experiments with combining their algorithms with both FixMatch and ReMixMatch).

The proposed algorithm make class balanced validation set from the train set, which seems not realistic. For example, for the iNaturalist dataset, the most minor class has only 2 samples, and in this case, it would be hard to make validation set from the 2 samples.

**Questions:**

I have no question.

---

> ### Author Response · Authors · 2023-11-19
> **Response to Reviewer PXgw**
>
> > Q1. There is no theoretical ground for the proposed algorithm.
>
> Now we rephrase our method section and supply SEVAL with its theoretical motivation. We also derive the principal hypothesis behind the success of SEVAL. Like other SSL algorithms, SEVAL is mostly heuristic. However, we believe it is valuable as our analysis sheds new lights on the problem of pseudo-label refinement and we validate the effectiveness with extensive experiments.
>
> > Q2. For CIFAR-10, the experimental settings are not enough. For example, other LTSSL (long tailed semi supervised learning) studies such as DARP, DASO, SAW conducted experiments on CIFAR-10 with changing both imbalanced ratio of labeled and unlabeled sets.
>
> Due to page limitations, we only report key experiment results in main texts. We provide plenty of experiments in section B. For the experiments of CIFAR10-LT, now we provide the results when $\gamma_l \neq \gamma_u$ and $\gamma_l=\gamma_u=150$. Take this into account, our experiments of CIFAR10-LT are equally or more extensive than previous works including DARP, DASO and SAW.
>
> > Q3. The proposed algorithm appears to require extensive hyperparameter tuning.
>
> SEVAL requires three core hyperparameters, including $t$ and two moments for curriculum learning ($\rho_\pi $ and $ \rho_\tau$). Other parameters, such as the length of the acquired curriculum ($L$), predominantly affect SEVAL's learning speed with minimal influence on the outcomes. Based on empirical observations, we reduce $L$ as the number of classes increases to enhance training efficiency.
>
> When compared with FixMatch, we only add two hyper-parameters because $t$ is also required for FixMatch. Considering that it is reasonable to keep $\rho_\pi $ and $\rho_\pi $ the same, we believe that SEVAL does not require extensive hyper-parameter tuning.
>
> Now we conduct sensitivity analysis of these parameters in section B.6 and we find SEVAL is not sensitive to hyper-parameters. We summarize the detailed hyper-parameter in section C.4 to facilitate reproducibility.
>
>
> > Q4. The proposed algorithm and baseline algorithms are combined with only FixMatch, without being combined with ReMixMatch (other LTSSL studies such as DARP, ABC, SAW and CoSSL conducted experiments with combining their algorithms with both FixMatch and ReMixMatch).
>
> We think there might be some misunderstandings. We integrate SEVAL with Mean Teacher, MixMatch and ReMixMatch and report experimental results in Figure 4(a), showing better performance than other methods. Now we report more detailed and complete results in section B.4 and summarize implementation details in section C.4.
>
> > Q5. ­­­The proposed algorithm make class balanced validation set from the train set, which seems not realistic. For example, for the iNaturalist dataset, the most minor class­ has only 2 samples, and in this case, it would be hard to make validation set from the 2 samples.
>
> We also think there might be some misunderstandings. As we demonstrate in the beginning of section 4, we make no assumption regarding the distribution of the validation set. To achieve so, we make our optimization function class-balanced, c.f. Eq.3 and Eq. 4. In fact, as we describe in the method section, we partition the training dataset into two subset with the same number of samples. Therefore, in most of our experiments, the validation set is also (highly) imbalanced.
>
> Regarding sample efficiency, we validate our algorithms using Semi-Aves, derived from the mentioned iNaturelist dataset. We find SEVAL works well under such settings and summarize the results in section B.2. SEVAL achieves this with frequency normalization and group-wise optimization, as mentioned in section 4.2 and described in details in section C. Now we also supply extreme experiment settings when very few labelled samples are available (tailed class contains 2 labelled samples or 40 labelled samples in total) in section B.3. We find SEVAL works well under such scenarios, demonstrating that SEVAL is labelled sample efficient.
>
>
> *We kindly ask the reviewer to reassess the merit of our work, considering both our technical contributions and the effectiveness of our method.*

---

> > ### Comment · Reviewer_PXgw · 2023-11-19
> > **Thank you for the response**
> >
> > Your response has addressed many of my concerns. Therefore, I would like to raise my score from 3 to 4.
> >
> > However, since there is no score of 4 in the ICLR scoring system, I will keep it as 3 for now. It would be appreciated if the meta reviewer could consider reflecting my score as 4 in the final decision.

---

### Official Review · Reviewer_axcv · 2023-10-29

**Soundness:** 2 fair
**Presentation:** 3 good
**Contribution:** 2 fair
**Rating:** 3
**Confidence:** 5

**Summary:**

This work studies the imbalance issue in semi-supervised learning. They develop a curriculum for adjusting logits before generating pseudo labels from biased models. Then, they build a curriculum for class-specific thresholds. They evaluate it on FixMatch over several datasets and show the results.

**Strengths:**

The paper is well-structured and easy to follow. The method is clearly presented. The algorithm can be useful for deploying semi-supervised learning in real-world scenarios.

**Weaknesses:**

1. Lack of technical novelty. All technical components have been established in the prior works, such as FlexMatch [1], Dash [2]. Specifically, Sec 4.1 -> [3]; Sec 4.2 -> [2, 4]; Sec 4.3 -> [1].
2. Lack of literature review. This work missed a couple of important baselines to compare with. For example, the methods presented in [4,5,6,7, 8].
3. Lack of experimental design. This work only evaluated it on FixMatch. However, in other pieces of literature [3,5,6,8], they evaluated on various SSL algorithms to show the effectiveness of the method, such as MixMatch and ReMixMatch. It would be helpful if the authors can present the results of these SSL algorithms. Moreover, FixMatch was proposed three years ago. It would be interesting to apply SEVAL to more recent algorithms, such as FlexMatch, CoMatch, or other SSL algorithms.
4. The motivation is not clear to me. The main goal of SSL is to reduce the labeling effort. However, the experimental setting shows they still use a large amount of labeled data in the training data and a labeled validation set. FixMatch can achieve ~90% accuracy when only using 40 labeled data. It would be helpful if the authors could do experiments where the labeled data are scarce, e.g., 40 labeled data in total, while the unlabeled data can be highly imbalanced.
5. The parameter is tuned on the validation set, which is not practical in my view. How do we prepare a labeled validation set in real-world scenario? If we have such labeling budget, why don't we choose other learning algorithms but stick to SSL?


[1] Zhang, Bowen, Yidong Wang, Wenxin Hou, Hao Wu, Jindong Wang, Manabu Okumura, and Takahiro Shinozaki. "Flexmatch: Boosting semi-supervised learning with curriculum pseudo labeling." Advances in Neural Information Processing Systems 34 (2021): 18408-18419.

[2] Xu, Yi, Lei Shang, Jinxing Ye, Qi Qian, Yu-Feng Li, Baigui Sun, Hao Li, and Rong Jin. "Dash: Semi-supervised learning with dynamic thresholding." In International Conference on Machine Learning, pp. 11525-11536. PMLR, 2021.

[3] Lai, Zhengfeng, Chao Wang, Sen-ching Cheung, and Chen-Nee Chuah. "SAR: Self-adaptive refinement on pseudo labels for multiclass-imbalanced semi-supervised learning." In Proceedings of the IEEE/CVF Conference on Computer Vision and Pattern Recognition, pp. 4091-4100. 2022.

[4] Guo, Lan-Zhe, and Yu-Feng Li. "Class-imbalanced semi-supervised learning with adaptive thresholding." In International Conference on Machine Learning, pp. 8082-8094. PMLR, 2022.

[5] Chen, Hao, Yue Fan, Yidong Wang, Jindong Wang, Bernt Schiele, Xing Xie, Marios Savvides, and Bhiksha Raj. "An Embarrassingly Simple Baseline for Imbalanced Semi-Supervised Learning." arXiv preprint arXiv:2211.11086 (2022).

[6] Lai, Zhengfeng, Chao Wang, Henrry Gunawan, Sen-Ching S. Cheung, and Chen-Nee Chuah. "Smoothed adaptive weighting for imbalanced semi-supervised learning: Improve reliability against unknown distribution data." In International Conference on Machine Learning, pp. 11828-11843. PMLR, 2022.

[7] Wei, Tong, and Kai Gan. "Towards Realistic Long-Tailed Semi-Supervised Learning: Consistency Is All You Need." In Proceedings of the IEEE/CVF Conference on Computer Vision and Pattern Recognition, pp. 3469-3478. 2023.

[8] Kim, Jaehyung, Youngbum Hur, Sejun Park, Eunho Yang, Sung Ju Hwang, and Jinwoo Shin. "Distribution aligning refinery of pseudo-label for imbalanced semi-supervised learning." Advances in neural information processing systems 33 (2020): 14567-14579.

**Questions:**

See weaknesses.

---

> ### Author Response · Authors · 2023-11-19
> **Response to Reviewer axcv (Part 1/2)**
>
> > Q1. Lack of technical novelty. All technical components have been established in the prior works, such as FlexMatch [1], Dash [2]. Specifically, Sec 4.1 -> [3]; Sec 4.2 -> [2, 4]; Sec 4.3 -> [1].
> [1] Zhang, Bowen, Yidong Wang, Wenxin Hou, Hao Wu, Jindong Wang, Manabu Okumura, and Takahiro Shinozaki. "Flexmatch: Boosting semi-supervised learning with curriculum pseudo labeling." Advances in Neural Information Processing Systems 34 (2021): 18408-18419.
> [2] Xu, Yi, Lei Shang, Jinxing Ye, Qi Qian, Yu-Feng Li, Baigui Sun, Hao Li, and Rong Jin. "Dash: Semi-supervised learning with dynamic thresholding." In International Conference on Machine Learning, pp. 11525-11536. PMLR, 2021.
> [3] Lai, Zhengfeng, Chao Wang, Sen-ching Cheung, and Chen-Nee Chuah. "SAR: Self-adaptive refinement on pseudo labels for multiclass-imbalanced semi-supervised learning." In Proceedings of the IEEE/CVF Conference on Computer Vision and Pattern Recognition, pp. 4091-4100. 2022.
> [4] Guo, Lan-Zhe, and Yu-Feng Li. "Class-imbalanced semi-supervised learning with adaptive thresholding." In International Conference on Machine Learning, pp. 8082-8094. PMLR, 2022.
> [5] Chen, Hao, Yue Fan, Yidong Wang, Jindong Wang, Bernt Schiele, Xing Xie, Marios Savvides, and Bhiksha Raj. "An Embarrassingly Simple Baseline for Imbalanced Semi-Supervised Learning." arXiv preprint arXiv:2211.11086 (2022).
> [6] Lai, Zhengfeng, Chao Wang, Henrry Gunawan, Sen-Ching S. Cheung, and Chen-Nee Chuah. "Smoothed adaptive weighting for imbalanced semi-supervised learning: Improve reliability against unknown distribution data." In International Conference on Machine Learning, pp. 11828-11843. PMLR, 2022.
> [7] Wei, Tong, and Kai Gan. "Towards Realistic Long-Tailed Semi-Supervised Learning: Consistency Is All You Need." In Proceedings of the IEEE/CVF Conference on Computer Vision and Pattern Recognition, pp. 3469-3478. 2023.
> [8] Kim, Jaehyung, Youngbum Hur, Sejun Park, Eunho Yang, Sung Ju Hwang, and Jinwoo Shin. "Distribution aligning refinery of pseudo-label for imbalanced semi-supervised learning." Advances in neural information processing systems 33 (2020): 14567-14579.
>
> We appreciate the reviewers for providing valuable insights from the literature. We do discuss and compare with [1, 2, 4, 5, 7, 8] in the manuscript and now add the discussion of [3, 6] in section 2. In particular, now we clearly show the differences between SEVAL and its counterparts either based on pseudo-label refinement or threshold adjustment in section 4.
>
> We would like to underscore that our principal contribution centers around systematically reviewing and enhancing existing methods within a unified framework. We observe that these methods are suboptimal due to issues related to either uncalibrated confidence or unreliable difficulty estimation. We elucidate our discoveries using Bayes' theorem and illustrate them through the two-moon example. This provides novel insights into the challenges of imbalanced semi-supervised learning and guides the development of SEVAL, a straightforward yet effective technique for optimization.
>
> The derived method, SEVAL, although share some similarity with the methods the reviewers mentioned (as they all refine pseudo-label or/and adjust thresholds), has some unique properties including building a theoretically more accurate classifier for unlabelled data (c.f. section 4.1) and accommodate more thresholding scenarios (c.f. section 4.2). We prove the feasibility of learning a curriculum of label refinement and thresholding parameters with a separate learning process using a partition of training dataset (c.f section 4.3). We think the technical contributions are therefore substantial. The extensive experiments further support the efficacy of SEVAL.
>
> > Q2. Lack of literature review. This work missed a couple of important baselines to compare with. For example, the methods presented in [4,5,6,7, 8].
>
> We think there might be some misunderstandings. We compare with Adsh [4], ACR [7], DARP [8] and report the results under the same setting mainly in Table 2 and also other tables. We discuss their differences with SEVAL in related work and method sections. We also mention [5] in related work, and compare with a very similar work (CreST) along most tables. Now we add a more detailed discussion of SAW [7] in section 2, which focuses on reweighting of unlabelled samples, therefore is orthogonal to our studies which focus on pseudo-label refinement. We think SEVAL and SAW can work well together and leave the investigation for future works.

---

> > ### Author Response · Authors · 2023-11-19
> > **Response to Reviewer axcv (Part 2/2)**
> >
> > > Q3. Lack of experimental design. This work only evaluated it on FixMatch. However, in other pieces of literature [3,5,6,8], they evaluated on various SSL algorithms to show the effectiveness of the method, such as MixMatch and ReMixMatch. It would be helpful if the authors can present the results of these SSL algorithms. Moreover, FixMatch was proposed three years ago. It would be interesting to apply SEVAL to more recent algorithms, such as FlexMatch, CoMatch, or other SSL algorithms.
> >
> > We think there might be some misunderstandings. We integrate SEVAL with Mean Teacher, MixMatch, and ReMixMatch, reporting experimental results in Figure 4(a), demonstrating superior performance compared to other methods. Now we report more detailed and complete results in section B.4 and summarize implementation details in section C.4.
> >
> > Our method cannot apply to FlexMatch as they both modify the thresholding. Instead, we conduct a thorough comparison between SEVAL and FlexMatch throughout this manuscript. CoMatch enhances the pseudo-label through feature regularization, a technique akin to the semantic alignment regularization introduced by DASO. We report our method and work well with semantic alignment regularization in section 5.2.1. We anticipate SEVAL can also work well with CoMatch, and we would like to leave these investigations for the future.
> >
> > > Q4. The motivation is not clear to me. The main goal of SSL is to reduce the labeling effort. However, the experimental setting shows they still use a large amount of labeled data in the training data and a labeled validation set. FixMatch can achieve ~90% accuracy when only using 40 labeled data. It would be helpful if the authors could do experiments where the labeled data are scarce, e.g., 40 labeled data in total, while the unlabeled data can be highly imbalanced.
> >
> > We utilize labelled validation dataset to learn the curriculum parameters because we find this can lead to better label refinement and thresholding, further improving current SSL algorithms. Existing methods cannot attain this without optimization on the validation dataset, as we elucidate in Section 4.
> >
> > Following the suggestion of the reviewers, now we provide experiment results under extreme settings with very few labeled samples (2 labeled samples in the tailed class or 40/100 labeled samples in total) in Section B.3. Our findings indicate that SEVAL performs effectively in such scenarios, highlighting its efficiency with limited labeled samples.
> >
> > > Q5. The parameter is tuned on the validation set, which is not practical in my view.
> >
> > Tuning the hyper-parameter based on the performance of the validation dataset is a common practice to develop machine learning models.
> >
> > SEVAL necessitates three key hyper-parameters, which include $t$ and two moments for curriculum learning ($\rho_\pi$ and $\rho_\tau$). In comparison to FixMatch, we only introduce two additional hyper-parameters because $t$ is also required for FixMatch. Given the reasonable assumption of maintaining consistency between $\rho_\pi$ and $\rho_\pi$, we assert that SEVAL does not demand extensive hyper-parameter tuning.
> >
> > In Section B.6, now we conduct a sensitivity analysis of these parameters and observe that SEVAL exhibits robustness to hyper-parameter variations. Detailed hyper-parameter information is summarized in Section C.4 to facilitate reproducibility.
> >
> > > Q6. How do we prepare a labeled validation set in real-world scenario?
> >
> > We think there might be some misunderstandings. As we demonstrate in section 4.3, in practice SEVAL does not require additional labelled validation sets. Instead, we learn the curriculum based on a partition of the training dataset. Specifically, we partition the training dataset into two equal subsets and utilize one of them for SEVAL parameter optimization. We do not utilize any additional labelled samples compared with other SSL algorithms.
> >
> > > Q7. If we have such labeling budget, why don't we choose other learning algorithms but stick to SSL?
> >
> > SEVAL is labelled data efficient and does not require more labelled samples than other SSL algorithms. In all the experimental settings we investigate in, we observe that SEVAL consistently outperforms state-of-the-art SSL algorithms with the same labeled budgets, even when the number of labeled samples is extremely limited (e.g., 40 in total).
> >
> > *We kindly ask the reviewer to reconsider the value of our work within the context of decomposing and advancing the design of pseudo-label based SSL rather than mainly an alternative method for pseudo-label refinement.*

---

> ### Comment · Reviewer_axcv · 2023-11-19
> **Thank you for the response.**
>
> Your response has addressed many of my concerns. I would like to raise my score from 3 to 3.5. However, since there is no score of 3.5 in the ICLR scoring system, I will keep it as 3 for now. It would be appreciated if the meta reviewer could consider reflecting my score as 3.5 in the final decision.
>
>  However, the comparison is still not enough compared to the existing papers, especially to SimiS [5]. In [5], they did a comprehensively job in benchmarking all settings and algorithms in Table 1. I'm interested in the results of SEVAL under the settings of Table 1 & 3 in [5] to have a comprehensive comparison with previous methods.

---

### Official Review · Reviewer_TEBc · 2023-10-30

**Soundness:** 2 fair
**Presentation:** 2 fair
**Contribution:** 2 fair
**Rating:** 5
**Confidence:** 3

**Summary:**

The performance of pseudo label based semi-supervised learning (SSL) methods heavily depends on the quality of pseudo labels. In order to alleviate the adverse impact of class imbalance on the quality of pseudo labels, a class-specific calibration and a class-specific threshold are learned, thereby improving the performance of imbalanced SSL methods.

**Strengths:**

1. The proposed method is of universality and can be embedded into other SSL methods to improve their performance on imbalanced classification problems.
2. A large number of experiments are conducted to demonstrate the effectiveness of the proposed method.

**Weaknesses:**

1. The discussion of relevant work was not sufficient, so that the progressiveness and novelty of the article could not be highlighted. As mentioned in the experimental section, some existing methods also include both pseudo label refinement (PLR) and threshold adjustment (THA) modules. A detailed discussion is needed on the differences between the proposed method and these methods.
2. Relevant theoretical analysis is needed to ensure the rationality of the proposed method. Otherwise, the proposed method is more like a combination of techniques.
3. The expression of the article needs to be improved, such as
(a) grammar error, e.g., in Page 4, “It which comprise two...”
(b) spelling error, e.g., in Page 9, “We compared with the ... inf Figure 8.”
(c) Figure 9 is not referenced in the main text.
(d) inconsistent expression, e.g., $n_i $ and $n1$ are both appeared.
(e) improper use of terminology, e.g., in Page 3, the meaning of "data loader" is unclear.

**Questions:**

1. In experimental section, why not record specific evaluation criteria for imbalanced classification, such as precision, recall, and F1 score?
2. Is the accuracy recorded in the experiment evaluated on the unlabeled samples used during the training phase? Is it evaluated on a test set that is not used during the training phase?
3. In the experimental section, is the class distribution on the data set used in the test phase the same as the labeled sample set used in the training phase? In real applications, how to ensure that the class distribution on a small number of labeled samples is the same as the real class distribution?
4. In Table 1, why is the imbalanced ratio of unlabeled data set unknown on the STL10-LT?
5. In Table 1, what does $n_1$ mean? Is it the number of labeled samples from majority class or the number of labeled samples from minority class?
6. On line 16 of Algorithm 2, why not use the final $\pi$ and $\tau$ learned in the previous stage?

---

> ### Author Response · Authors · 2023-11-19
> **Response to Reviewer TEBc (Part 1/2)**
>
> > Q1. The discussion of relevant work was not sufficient, so that the progressiveness and novelty of the article could not be highlighted. As mentioned in the experimental section, some existing methods also include both pseudo label refinement (PLR) and threshold adjustment (THA) modules. A detailed discussion is needed on the differences between the proposed method and these methods.
>
> We have rephrased our method section and highlighted the differences between SEVAL and other methods focusing on PLR and THA. We demonstrate distinct variations from both a theoretical standpoint and through easily comprehensible toy examples. In the context of PLR, SEVAL can yield more optimal results by leveraging Bayes' theorem, as elaborated in section 4.1. In the case of THA, SEVAL stands out as the sole method that estimates thresholds based on precision, offering a more reliable approach compared to existing methods, as discussed in section 4.2.
>
> > Q2. Relevant theoretical analysis is needed to ensure the rationality of the proposed method. Otherwise, the proposed method is more like a combination of techniques.
>
> We have enhanced the method section, providing SEVAL with a robust theoretical foundation. Additionally, we have elucidated the core hypothesis underpinning SEVAL's effectiveness. Although SEVAL shares the heuristic nature typical of many SSL algorithms, its value lies in our analysis, offering novel insights into the pseudo-label refinement challenge. We substantiate SEVAL's efficacy through comprehensive experimental validation.
>
> > Q3. The expression of the article needs to be improved, such as (a) grammar error, e.g., in Page 4, “It which comprise two...” (b) spelling error, e.g., in Page 9, “We compared with the ... inf Figure 8.” (c) Figure 9 is not referenced in the main text. (d) inconsistent expression, e.g., n_{1} and n1 are both appeared. (e) improper use of terminology, e.g., in Page 3, the meaning of "data loader" is unclear.
>
> We thank the reviewer for detailed suggestions. All the identified typos have been rectified.
>
> > Q4. In experimental section, why not record specific evaluation criteria for imbalanced classification, such as precision, recall, and F1 score?
>
> We follow the literature and utilize accuracy as a main metric to evaluate the effectiveness of SSL algorithms. Of note, as the test data has a uniform distribution, the reported accuracy is equivalent to average recall of all classes and can represent the overall classification performance of all classes.
>
> We agree with the reviewers on the importance of comprehensive evaluation metrics. Due to page limitations, we summarize the class-wise precision, recall and F1 score in appendix section E. From the detailed results, we find that SEVAL improves the classification model by being more sensitive to the minority classes.
>
> > Q5. Is the accuracy recorded in the experiment evaluated on the unlabeled samples used during the training phase? Is it evaluated on a test set that is not used during the training phase?
>
> We report the accuracy on a test set that is not used during the training phase. We do not evaluate the performance on the unlabeled data as it is out of scope for the problem setting of SSL.

---

> ### Author Response · Authors · 2023-11-19
> **Response to Reviewer TEBc (Part 2/2)**
>
> > Q6. In the experimental section, is the class distribution on the data set used in the test phase the same as the labeled sample set used in the training phase?
>
> Following the literature, the distribution of test data is different from that of training data. In particular, the test data is assumed to follow a uniform distribution, because we care more about the overall classification performance.
>
> > Q7. In real applications, how to ensure that the class distribution on a small number of labeled samples is the same as the real class distribution?
>
> Our algorithms and experimental setting assume that the training data follows the long-tailed distributions, which is common for real world datasets. We make no assumption on the distribution of the validation dataset, as we declare in the beginning of section 4. This is achieved class-balanced optimization function, c.f. Eq.3 and Eq. 4.
>
> > Q8. In Table 1, why is the imbalanced ratio of unlabeled data set unknown on the STL10-LT?
>
> The purpose of STL10 is to make use of the unlabelled data, which comes from a similar but different distribution from the labelled data, to learn better representation. Therefore, this dataset does not provide any class prior of unlabelled  samples, making it impossible for us to control the imbalanced ratio.
>
> > Q9. In Table 1, what does n1 mean? Is it the number of labeled samples from majority class or the number of labeled samples from minority class?
>
> $n_{1}$ represents the samples from the class that contains the most samples as $n_{c}$ is sorted in a descending order. Now we further clarify this in section 3.
>
> > Q10. On line 16 of Algorithm 2, why not use the final and learned in the previous stage?
>
> On line 16 of Algorithm 2, we calculate the step of curriculum l. This is an index used to retrieve the learned parameters from the previous stage, i.e. line 18 and line 19.
>
> We refrain from relying on the ultimate learned results due to the dynamic nature of optimal parameters throughout the training process. The shifting model bias and evolving classification capability during learning necessitate an alternative approach. Thus, in section 4.3, we introduce curriculum learning, aiming to acquire a set of parameters that adapt to the evolving learning process.

---

### Official Review · Reviewer_S1He · 2023-10-31

**Soundness:** 3 good
**Presentation:** 2 fair
**Contribution:** 2 fair
**Rating:** 5
**Confidence:** 4

**Summary:**

This paper proposed a solution for threshold adjustment for pseduo-labeling based semi-supervised learning (SSL) method when there exists class-imbalance. The proposed solution leverages a seperately labeled validation set to optimize the thresholds, and can be applied as an add-on to existing SSL methods.

**Strengths:**

1. This paper identified an important yet understudied question - how to learn pseudo-label thresholds when there exists class-imbalance.

2. This paper proposed a novel threshold adjustment scheme that is suitable for label imbalanced scenarios.

3. The proposed method appears to have notable improvements over state-of-the-art baselines.

**Weaknesses:**

Major issues:

1. This paper mandates the strong assumption that there must exist a sufficient amount of labeled validation data per class (at least 10 per class), which could easily be violated under semi-supervised learning settings.

2. Highly relevant and overlapping method such as Adsh [1] is not included in the experiments.

Minor issues:

3. The motivation is not strong enough, in the abstract, the authors mentioned that existing dynamic threshold methods can cause bias in the pseudo-labeling. However, there seems a lack of detailed justification to substantiate how that method can cause bias. In addition, approaches such as FlexMatch and FreeMatch use class-dependent thresholds, shouldn't this approach reduce the bias caused by class imbalance?

4. The presentation could be further improved, specifically on the layout of Figures 3-5 and figure 7-9.

5. This paper is overall heuristic, the proposed solution lacks theoretical justification.

[1] Guo, Lan-Zhe, and Yu-Feng Li. "Class-imbalanced semi-supervised learning with adaptive thresholding." International Conference on Machine Learning. PMLR, 2022.

**Questions:**

1. The authors are encouraged to improve the proposed method in situations where the labeled set is not big enough.

2. I failed to conceive how curriculum learning reconciles with the threshold adjustment, is there a process where SEVAL reordering the data? Or the curriculum in this context is different from the commonly referred curriculum learning [1]?

[1] Bengio, Yoshua, et al. "Curriculum learning." Proceedings of the 26th annual international conference on machine learning. 2009.

---

> ### Author Response · Authors · 2023-11-19
> **Response to Reviewer S1He (Part 1/2)**
>
> > Q1. This paper mandates the strong assumption that there must exist a sufficient amount of labeled validation data per class (at least 10 per class), which could easily be violated under semi-supervised learning settings.
>
> SEVAL does not request more labelled samples than other SSL algorithms. Our ablation study demonstrates that SEVAL remains stable with varying amounts of validation data, ranging from 10 to 500 per class.
> In order to further demonstrate the data efficiency of SEVAL, now we also supply extreme experiment settings when very few labelled samples are available (tailed class contains 2 labelled samples or 40 labelled samples in total) in section B.3. The experimental results demonstrate that SEVAL is labelled sample efficient and works well under such scenarios.
>
> > Q2. Highly relevant and overlapping method such as Adsh [1] is not included in the experiments.
> [1] Guo, Lan-Zhe, and Yu-Feng Li. "Class-imbalanced semi-supervised learning with adaptive thresholding." International Conference on Machine Learning. PMLR, 2022.
>
> Following the request of the reviewers, now we add the comparison with Adsh in the experiments. We now update the results in Table.2. We find that Adsh improves model performance in most cases. Our algorithm is consistently better than Adsh in all the settings.
>
> > Q3. The motivation is not strong enough, in the abstract, the authors mentioned that existing dynamic threshold methods can cause bias in the pseudo-labeling. However, there seems a lack of detailed justification to substantiate how that method can cause bias.
>
> The bias in pseudo-labeling arises from disparities in class distributions between training and test data. As discussed in Theorem 1, the optimal decision boundary for test data should consider differences in marginal distributions. Current solutions fall short, either by neglecting the class prior of test data or lacking calibrated probabilities, as illustrated in section 4.1.
> We have emphasized the reason for class bias and the problems of current methods in section 4.1.
>
> > Q4: In addition, approaches such as FlexMatch and FreeMatch use class-dependent thresholds, shouldn't this approach reduce the bias caused by class imbalance?
>
> Current dynamic threshold methods (including FlexMatch and FreeMatch) mostly adjust thresholding based on maximum class probability (MCP), which is an approximation of recall per class. However, as we analyze in Section 4.2, the optimal thresholds should correlate with precision rather than recall. Therefore, as we demonstrate in a two-moon toy example, existing MCP-based methods would fail two out of the four threshold adjustment scenarios, therefore not optimal. In contrast, SEVAL can accommodate all the four cases. We also validate this in CIFAR10-LT experiments in section D. Moreover, we  show that the selected pseudo-labels are more correct than FlexMatch and FreeMatch in Figure 3(a).
> We have rephrased the method section to emphasize these differences with more analysis and evidence.

---

> > ### Author Response · Authors · 2023-11-19
> > **Response to Reviewer S1He (Part 2/2)**
> >
> > > Q5. The presentation could be further improved, specifically on the layout of Figures 3-5 and figure 7-9.
> >
> > We thank the reviewers for the suggestions. We now polish the layout of all the figures, including the mentioned two (now Figure 3 and Figure 4).
> >
> > > Q6. This paper is overall heuristic, the proposed solution lacks theoretical justification.
> >
> > We have revised the method section and provided SEVAL with its theoretical foundation. Additionally, we have formulated the key hypothesis underpinning the success of SEVAL. While SEVAL, like other SSL algorithms, is primarily heuristic, we consider it valuable as our analysis offers new insights into the pseudo-label refinement problem, and we support the effectiveness of SEVAL through comprehensive experiments.
> >
> > > Q7. The authors are encouraged to improve the proposed method in situations where the labeled set is not big enough.
> >
> > As also discussed in Q1, SEVAL works well when the labelled set is small. In particular, SEVAL accomplishes this through frequency normalization and group-wise optimization, as described in section 4.2 and detailed in Section C.
> >
> > > Q8. I failed to conceive how curriculum learning reconciles with the threshold adjustment, is there a process where SEVAL reordering the data? Or the curriculum in this context is different from the commonly referred curriculum learning [1]?
> > [1] Bengio, Yoshua, et al. "Curriculum learning." Proceedings of the 26th annual international conference on machine learning. 2009.
> >
> > SEVAL does not exactly ‘rearrange’ the data but instead organizes the learning process in a manner that incrementally raises the level of difficulty. This is achieved by the two curriculum parameters $\boldsymbol{\pi}$ and $\boldsymbol{\tau}$. Specifically, $\boldsymbol{\pi}$ will gradually reduce the class bias of the minority class while $\boldsymbol{\tau}$ will gradually include more samples for training. This is connected to the concept commonly known as curriculum learning, a term used similarly in related studies, such as in the case of FlexMatch.

---

### Author Response · Authors · 2023-11-19
**General response and summary of changes**

We thank all reviewers for their time and feedback. Reviewers point out the addressed problem is important (Reviewer S1He), the approaches are novel (Reviewer S1He) and of universality (Reviewer TEBc), evaluation is thorough (Reviewer TEBc), the improvements are significant (Reviewer S1He) and the presentation is clear (Reviewer axcv). As highlighted by Reviewer S1He, it investigates how to learn pseudo-label thresholds when there exists class imbalance, to enable semi-supervised learning in real world applications, and highlights the importance of learning label refinement and thresholds in a unified framework.

In addition to the point-to-point responses to reviewers, we would like to emphasize the clarifications of two main concerns to resolve misunderstanding.

1. Technical novelty of learning label refinement and thresholds. As our main contribution, we review existing methods with a unified framework and find that they are not optimal because of either uncalibrated probability or unreliable difficulty estimation (**which we now emphasize more in the method section**). We explain our findings with Bayes’ theorem and demonstrate with a two-moon example. This sheds new lights on imbalance SSL and helps us derive SEVAL, which is a simple yet effective technique to achieve optimization.
2. SEVAL is (labeled) data efficient. Although SEVAL learns from a partition of training dataset, **it does not require more labelled samples than any other SSL algorithms**. SEVAL achieves this with frequency normalization and group-wise optimization, as we highlight in section 4.2 and detail in section C. We demonstrate this with numerous experiments, where SEVAL only accesses the same amount of labeled training samples and outperforms its counterparts. During rebuttal, we conduct stress tests on SEVAL and observe its effectiveness, even with only 40 labelled data points in total, or 2 labelled data in tail class.


We are grateful for the valuable feedback provided by the reviewers, and we have integrated it into the updated version of the manuscript. We marked the modifications with $\textcolor{red}{red}$ color in the revised manuscript. To sum up, the primary modifications made during this rebuttal period include:

* We try to make the theoretical motivation of SEVAL crystal clear by rephrasing most parts of section 4. We enrich the method description with more theoretical analysis. We also illustrate the effectiveness of our methods with two-moon toy experiments. We believe this can help highlight our technical contributions.
* We supply experimental results when SEVAL is trained with very few labelled samples and summarize the results in section B.3. We believe this can help address the concerns of SEVAL data efficiency.
* We extend SEVAL experiments based on SSL algorithms other than FixMatch and summarize the implementation details in section C.5 and results in section B.5. We believe this can further validate SEVAL’s flexibility.
* We extend experiments on CIFAR10-LT with varied imbalanced ratios and summarize the results in section B.1. We believe this can further validate the effectiveness of SEVAL.
* We conduct sensitive analysis of SEVAL hyper-parameters and summarize the results in section B.4. We believe this can help the practitioners quickly adopt SEVAL to their tasks at hand.
* We report fine-grained class-wise performance of recall, precision, and F1-score in section E. We believe this can help the readers access the improvements better.
* We add a relevant baseline method of Adsh in Table 2. We hope this can help the readers access the effectiveness of SEVAL.
* We add discussion of two relevant methods including SAW and SaR in section 2. We hope this can help the readers understand the technical contributions of SEVAL better.
* We fix several typos and polish all figure layouts. We hope this can make our manuscript clearer and easier to follow.

---

### Meta-Review · Area_Chair_Wpfo · 2023-12-07

**Metareview:**

This paper studies the imbalance problem in semi-supervised learning. The authors develop a strategy for adjusting logits before generating pseudo labels from biased models. Then, they build a curriculum for class-specific thresholds. The experiments show that SEVAL surpasses current methods based on pseudo-label refinement and threshold adjustment.

Most of the reviewers acknowledge the strengths in the writing and experimental aspects of the paper. However, they also point out several weaknesses: (1) the method is considered incremental based on existing methods; (2) the motivation is not clear; (3) the discussion of related work is not comprehensive. Based on the overall reviews received for this paper, I am inclined to reject this paper.

**Justification For Why Not Higher Score:**

1. The method is considered incremental based on existing methods such as Dash [2] and Flexmatch [1]. The proposed method depends on a strong assumption that there must exist a sufficient amount of labeled validation data per class (at least 10 per class), which could easily be violated under semi-supervised learning settings.

2. The motivation is not strong enough. There seems a lack of detailed justification to substantiate how that method can cause bias. Moreover, the proposed algorithm appears to require extensive hyperparameter tuning to reduce the bias.

3. Some references and experimental comparisons are missed.

[1] Zhang, Bowen, Yidong Wang, Wenxin Hou, Hao Wu, Jindong Wang, Manabu Okumura, and Takahiro Shinozaki. "Flexmatch: Boosting semi-supervised learning with curriculum pseudo labeling." Advances in Neural Information Processing Systems 34 (2021): 18408-18419.

[2] Xu, Yi, Lei Shang, Jinxing Ye, Qi Qian, Yu-Feng Li, Baigui Sun, Hao Li, and Rong Jin. "Dash: Semi-supervised learning with dynamic thresholding." In International Conference on Machine Learning, pp. 11525-11536. PMLR, 2021.

**Justification For Why Not Lower Score:**

N/A.

---

### Decision · Program_Chairs · 2024-01-16

Reject